# STRICT SUBGOAL EXECUTION: RELIABLE LONG-HORIZON PLANNING IN HIERARCHICAL REINFORCEMENT LEARNING

**Jaebak Hwang, Sanghyeon Lee, Jeongmo Kim, Seungyul Han**[*]
Graduate School of Artificial Intelligence
Ulsan National Institute of Science and Technology (UNIST)
Ulsan, South Korea 44919
`{milk, sanghyeon, jmkim22, syhan}@unist.ac.kr`

## ABSTRACT

Long-horizon goal-conditioned tasks pose fundamental challenges for reinforcement learning (RL), particularly when goals are distant and rewards are sparse. While hierarchical and graph-based methods offer partial solutions, their reliance on conventional hindsight relabeling often fails to correct subgoal infeasibility, leading to inefficient high-level planning. To address this, we propose Strict Subgoal Execution (SSE), a graph-based hierarchical RL framework that integrates Frontier Experience Replay (FER) to separate unreachable from admissible subgoals and streamline high-level decision making. FER delineates the reachability frontier using failure and partial-success transitions, which identifies unreliable subgoals, increases subgoal reliability, and reduces unnecessary high-level decisions. Additionally, SSE employs a decoupled exploration policy to cover underexplored regions of the goal space and a path refinement that adjusts edge costs using observed low-level failures. Experimental results across diverse long-horizon benchmarks show that SSE consistently outperforms existing goal-conditioned and hierarchical RL methods in both efficiency and success rate. Our code is available at `https://github.com/Jaebak1996/SSE`

## 1 INTRODUCTION

Recent advances in reinforcement learning (RL) have achieved impressive success across various domains (Mnih et al., 2013; Silver et al., 2016). Although prior works have enhanced sample efficiency through advanced exploration (Burda et al., 2018; Han & Sung, 2021a;b; Jo et al., 2024) and experience replay mechanisms (Schaul et al., 2015b; Han & Sung, 2017), they often struggle to adapt to changing objectives without extensive retraining. Consequently, Goal-Conditioned RL (GCRL) was introduced to explicitly condition the policy on a target goal, allowing agents to directly pursue diverse states specified by the environment (Schaul et al., 2015a; Levy et al., 2017; Nasiriany et al., 2019). However, in sparse-reward and long-horizon environments, goals are often distant, making exploration difficult and hindering effective learning due to a lack of intermediate guidance.

Hierarchical reinforcement learning (HRL) addresses long-horizon goal-conditioned tasks by decomposing control into a high-level policy that proposes subgoals or skills and a low-level policy that executes actions to reach them (Bacon et al., 2017; Vezhnevets et al., 2017). While this decomposition can ease learning by introducing more attainable intermediate objectives, performance can degrade when proposed subgoals are not reliably reachable by the low-level policy. To mitigate this issue, graph-based HRL methods construct a graph over the goal space where nodes represent states or regions and edges encode feasible transitions (Zhang et al., 2018; Nachum et al., 2018a; Huang et al., 2019; Eysenbach et al., 2019; Kim et al., 2021; Zhang et al., 2021). This structure enables subgoal selection via shortest-path planning. Despite these advances, scalability remains challenging when the final goal is far from the current state because high-level planning may require many

---

[*]Correspondence to: Seungyul Han

decisions and lead to unstable training. To reduce reliance on high-level policies, recent work has explored graph-based GCRL without explicit hierarchical structures (Lee et al., 2023; Yoon et al., 2024). However, removing high-level reasoning can reduce flexibility in environments with multiple goals or heterogeneous reward signals.

To overcome these limitations, we propose **Strict Subgoal Execution (SSE)**, a graph-based HRL framework that improves long-horizon performance by explicitly characterizing the reachability boundary of candidate subgoals. SSE integrates *Frontier Experience Replay* (FER) to distinguish high-level outcomes into failure transitions and partial-success transitions, localizing where execution breaks down and how far it progresses toward a subgoal. This frontier-based separation enables the agent to identify unreliable subgoals, train policies to execute admissible subgoals more accurately, and reduce unnecessary high-level decisions. SSE also expands goal-space coverage by actively exploring undervisited graph regions and improves planning reliability by refining paths to avoid transitions with frequent failures. As a result, SSE solves a broader range of complex long-horizon tasks. Our contributions are summarized as follows:

1. **Frontier Experience Replay (FER) for SSE:** We introduce FER, which delineates the reachability frontier using two high-level samples: failure transitions and partial-success transitions. By precisely localizing where attempts fail and how far progress extends, FER identifies unreliable subgoals, increases subgoal reliability and reduces unnecessary high-level decisions.

2. **Decoupled Exploration for Goal Space Coverage:** A dedicated exploration policy is decoupled from the return-driven high-level policy to traverse underexplored regions of the goal space, improving coverage and sample efficiency.

3. **Failure-Aware Path Refinement:** To improve subgoal reliability, we adjust graph edge costs based on low-level failure statistics, encouraging path planning to avoid unstable transitions and strengthening subgoal execution.

Through extensive evaluation on diverse long-horizon goal-conditioned tasks, our method demonstrates higher success rates and better sample efficiency than prior GCRL and HRL approaches, validating the effectiveness of the proposed SSE framework.

## 2 Preliminaries

### 2.1 Universal MDP, Goal-Conditioned RL, and Goal Relabeling Techniques

We consider a universal Markov decision process (UMDP) $(\mathcal{S}, \mathcal{G}, \mathcal{A}, P, R, \gamma)$, where $\mathcal{S}$, $\mathcal{G}$, and $\mathcal{A}$ denote the state, goal, and action spaces, $P$ the transition dynamics, $R$ the reward function, and $\gamma \in (0, 1]$ the discount factor Schaul et al. (2015a). At time step $t$, the agent observes $(s_t, g)$, samples $a_t \sim \pi(\cdot|s_t, g)$, and receives $r_t = R(s_t, a_t, s_{t+1}, g)$ with $s_{t+1} \sim P(\cdot|s_t, a_t)$. The goal of GCRL is to learn a goal-conditioned policy $\pi$ that maximizes the expected return $\sum_{t=0}^{H} r_t$ over horizon $H$. When $\mathcal{S} \neq \mathcal{G}$, GCRL typically assumes a mapping $\phi : \mathcal{S} \rightarrow \mathcal{G}$ projecting states into the goal space. The goal $g$ may be fixed or sampled per episode. In long-horizon goal-conditioned settings, learning is often inefficient due to sparse positive signals. Goal relabeling methods such as Hindsight Experience Replay (HER) (Andrychowicz et al., 2017) improve sample efficiency by treating later achieved states as substitute goals. HER replaces the goal $g$ with a future achieved goal $g' = \phi(s_{t'})$ for some $t' \geq t$ and recomputes the reward:

$$(s_t, a_t, r_t, s_{t+1}, g) \mapsto (s_t, a_t, R(s_t, a_t, s_{t+1}, g'), s_{t+1}, g').$$

This converts unsuccessful attempts into useful signals while the original transitions unchanged.

### 2.2 HRL Frameworks and Graph-Based Subgoal Planning in GCRL

In goal-conditioned settings, HRL addresses long-horizon challenges by decomposing control into a high-level policy $\pi^h$ and a low-level policy $\pi^l$ Bacon et al. (2017); Vezhnevets et al. (2017). In general, HRL frameworks in the GCRL domain typically assume that the goal space $\mathcal{G}$ is known to facilitate the high-level selection of feasible subgoals. Every $k$ steps, $\pi^h(\cdot \mid s_t, g)$ selects a subgoal $\tilde{g}_t \in \mathcal{G}$, which $\pi^l(\cdot \mid s_t, \tilde{g}_t)$ attempts to reach using an auxiliary reward (Zhang et al., 2020; Pateria et al., 2021; Hutsebaut-Buysse et al., 2022). However, when subgoals are too distant, the low-level policy may fail to reach them within the given horizon, and distance-based penalties can

hinder learning under sparse rewards. To mitigate this, graph-based approaches construct a goal-space graph $G = (V, E)$, where $V$ is a set of landmark nodes and $E$ contains edges weighted by the effort to transition between nodes. Landmarks are commonly chosen via farthest point sampling (FPS) (Kim et al., 2021; Lee et al., 2022) or placed on a predefined grid over the goal space (Yoon et al., 2024), assuming that the goal space is known to define the graph and the high-level policy.

In general, the edge cost is defined as

$$d(v_1 \rightarrow v_2) := \log_\gamma(1 + (1 - \gamma)Q^G(v_1, v_2, \pi^l)), \tag{1}$$

where $Q^G$ estimates the feasibility of reaching $v_2$ from $v_1$ under $\pi^l$, defined as the value function $Q^l$ of the low-level policy or a predefined estimator. In this paper, we set modified version of edge cost and details on $Q^G$ are provided in Appendix B.1. Given a subgoal $\tilde{g}_t$, graph-based HRL computes the shortest path from $\phi(s_t)$ to $\tilde{g}_t$ via Dijkstra's algorithm (Dijkstra, 1959), yielding waypoints $(\text{wp}_1, \ldots, \text{wp}_n)$ with $\text{wp}_i \in V$. The low-level policy sequentially targets these waypoints, switching to $\text{wp}_{i+1}$ once $\text{wp}_i$ is reached, until $\tilde{g}_t$ is attained. The high-level policy is trained on transitions $(s_t, g, \tilde{g}_t, \sum_{j=t}^{t+k-1} r_j, s_{t+k})$ in $\mathcal{B}_F^h$, while $\pi^l$ is trained on $(s_t, \text{wp}_i, a_t, r_t^l, s_{t+1})$ in $\mathcal{B}^l$. Following prior work (Kim et al., 2021; Lee et al., 2022), the low-level reward is $r_t^l = 0$ if $||\phi(s_{t+1}) - \text{wp}_i|| < \lambda$ and $-1$ otherwise, where $\lambda > 0$ is a subgoal reachability threshold and $\phi$ is known. In contrast, some approaches omit high-level subgoals and directly train the low-level policy to reach the final goal $g$, assuming intermediate subgoals are unnecessary (Lee et al., 2023; Yoon et al., 2024).

## 3 RELATED WORK

**Goal-Conditioned RL and Hierarchical Approaches**   GCRL refers to RL settings where the agent is explicitly conditioned on a goal (Kaelbling, 1993; Liu et al., 2022; Colas et al., 2022). Modern GCRL typically employs Universal Value Function Approximators (UVFA) (Schaul et al., 2015a) to generalize across goals. A central challenge is solving long-horizon tasks with sparse rewards, where exploration is difficult. To address this, HRL introduces multi-level policies that decompose complex tasks into temporally abstract subgoals (Vezhnevets et al., 2017; Nachum et al., 2018b) or reusable skills (Eysenbach et al., 2018; Lee et al., 2025), thereby reducing the effective planning horizon. The effectiveness of this decomposition has been demonstrated in diverse settings (Barto & Mahadevan, 2003; Kulkarni et al., 2016; Nachum et al., 2018a; 2019). Beyond hierarchical structures, other methods improve sample efficiency through structured exploration, such as prioritizing novel states (Warde-Farley et al., 2018; Pong et al., 2019) or discovering useful goals via unsupervised learning (Mendonca et al., 2021; Ecoffet et al., 2021; Chane-Sane et al., 2021).

**Relabeling and Guidance Techniques in GCRL**   In sparse-reward GCRL, data relabeling is central for improving sample efficiency. HER converts failed trajectories into successes by replacing the intended goal with a future achieved goal (Andrychowicz et al., 2017). Since uniform hindsight-goal sampling can be suboptimal, subsequent work explores curriculum-based relabeling (Fang et al., 2019), novelty- or priority-driven goal sampling (Zeng et al., 2023), and hierarchical relabeling that better matches low-level behavior (Nachum et al., 2018b). Orthogonal guidance approaches discourage infeasible subgoals, for example via learned adjacency constraints (Zhang et al., 2020) or adversarially generated goals that are challenging yet achievable (Levy et al., 2017).

**Graph-based Approaches in GCRL**   Graph-based approaches have been introduced to GCRL to provide a structured representation of the goal space, enabling planning over discrete landmarks and improving navigation in sparse-reward environments (Huang et al., 2019; Eysenbach et al., 2019; Kim et al., 2021). Early work used graph structures to represent the state space and to guide exploration (Zhang et al., 2018; Nachum et al., 2018a), and this direction has since evolved through integration with latent modeling (Zhang et al., 2021) and policy-driven graph construction (Kim et al., 2023). More recent advances focus either on aligning high-level decisions with low-level execution via graph-based planning (Lee et al., 2022) or on enhancing exploration with strategies such as frontier-based expansion (Hoang et al., 2021; Park et al., 2024), curriculum-based goal selection (Lee et al., 2023), and virtual subgoal generation for broader coverage (Yoon et al., 2024). In related works, *frontier-based exploration* refers to exploring near boundary states. In our work, we also use the term *frontier*, but it denotes the boundary that identifies failure points and separates achievable from unachievable subgoals, which is a distinct objective from frontier-based exploration.

# 4 METHODS

## 4.1 MOTIVATION: RETHINKING SUBGOAL EXECUTION IN GRAPH-BASED HRL

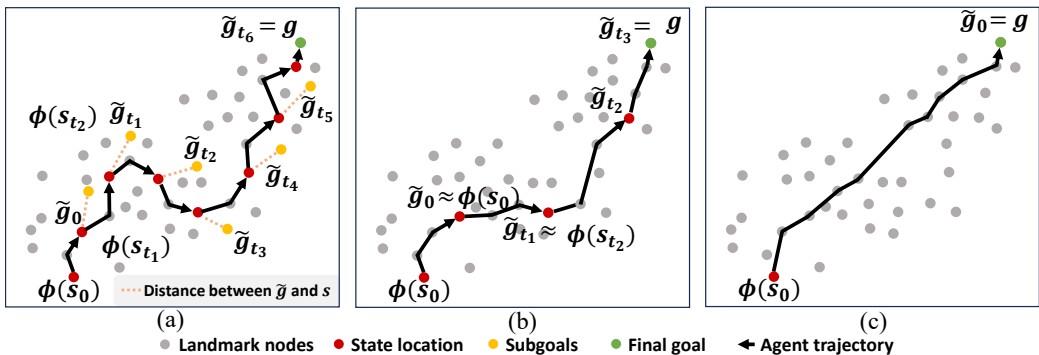

Figure 1: Agent trajectories in goal space $\mathcal{G}$. (a) Conventional HRL with HER relabels intermediate states as subgoals without enforcing exact subgoal completion, which lengthens high-level trajectories. (b) SSE with FER enforces exact subgoal completion, increasing subgoal reliability and reducing unnecessary high-level decisions, thereby improving learning efficiency. (c) After training, SSE reaches $g$ with few high-level steps, here in a single step in single-goal settings even from distant starts. Agent locations are $\phi(s_t) \in \mathcal{G}$ and $t_i$ is the $i$-th high-level step.

In GCRL, many methods improve goal-reaching performance by adopting hindsight relabeling techniques such as HER (Andrychowicz et al., 2017), which treat intermediate states as virtual goals to provide additional training signals. Recent graph-based HRL methods typically apply this idea to both the low level and the high level. While this benefits the low level, it introduces a critical issue at the high level. Fig. 1(a) illustrates how graph-based HRL with HER operates: the high-level policy selects a subgoal $\tilde{g}_t$ given the current state $s_t$, and the low-level policy follows a waypoint path to reach it. If $\tilde{g}_t$ is unreachable due to limited skill, obstacles, or excessive distance, the agent stops at an intermediate state, and such subgoals should be avoided. When HER is applied, every visited state along the failed trajectory is treated as if it were the intended subgoal. This causes the high-level policy to repeatedly select ineffective subgoals and to waste many decision steps. In addition, the resulting transitions can vary widely for the same subgoal, which hinders stable learning. As shown in Fig. 1(a), this produces unnecessarily long high-level trajectories, and even if the agent reaches the goal $g$, credit is spread over too many steps, preventing earlier decisions from being reinforced and often leading to failure on long-horizon tasks.

To address this issue, we propose the Strict Subgoal Execution (SSE) framework, which updates the high-level policy with positive returns only when the low level successfully reaches the assigned subgoal. We instantiate this principle with Frontier Experience Replay (FER). Unlike relabeling methods such as HER that synthesize additional successes, FER delineates the reachability frontier by recording two types of high-level samples, failure transitions and partial-success transitions. By precisely localizing where attempts fail and how far progress extends, FER identifies unreliable subgoals, provides consistent training signals, and reduces unnecessary high-level decisions. As shown in Fig. 1(b), this separation of success and failure keeps the resulting state $\phi(s_{t'})$ closely aligned with the selected subgoal $\tilde{g}_t$, producing consistent high-level transitions and eliminating wasteful actions. Consequently, as illustrated in Fig. 1(c), the resulting high-level policy solves tasks with far fewer decisions, often reaching the final goal $g$ in a single step in simple settings and handling multi-goal or long-horizon environments with only a few well-chosen subgoal selections.

## 4.2 STRICT SUBGOAL EXECUTION WITH FRONTIER EXPERIENCE REPLAY

In this section, we describe the details of the SSE framework. We first define FER, the key component of SSE, which marks the reachability frontier by recording two high-level sample types in addition to standard successes: failure transitions that stop at the point of failure and partial-success transitions that record the last reliably reached waypoint. To formalize this, we basically follow the HRL setup from Section 2: At each time $t$, the policy selects a subgoal $\tilde{g}_t \in \mathcal{G}$ under the common assumption that the goal space $\mathcal{G}$ is known. A waypoint path $(\mathrm{wp}_1, \ldots, \mathrm{wp}_n)$ on the graph $G = (V, E)$

is then generated, and the low-level policy $\pi^l$ follows this path until termination at time $t'$. We regard the subgoal as reachable if $\|\phi(s_{t'}) - \tilde{g}_t\| < \lambda$; otherwise the attempt is treated as a failure. In the failure case, $\mathrm{wp}_{\mathrm{final}}$ denotes the last waypoint reached within tolerance before failure. Based on this notion of reachability, we define FER as follows.

**Definition 4.1 (Frontier Experience Replay)** *The high-level replay buffer $\mathcal{B}_F^h$ stores transitions as*

$$
\mathcal{B}_F^h = \begin{cases}
(s_t, g, \tilde{g}_t, \sum_{j=t}^{t'-1} r_j, s_{t'}) \text{ (success)} & \text{if } \|\phi(s_{t'}) - \tilde{g}_t\| < \lambda, \\
(s_t, g, \tilde{g}_t, 0, s_T) \text{ (stop-on-failure)} & \text{if } \|\phi(s_{t'}) - \tilde{g}_t\| \geq \lambda, \\
(s_t, g, \mathrm{wp}_{\mathrm{final}}, \sum_{j=t}^{t_{\mathrm{wp}}-1} r_j, s_{t_{\mathrm{wp}}}) \text{ (partial success)} & \text{if failure occurs and } \mathrm{wp}_{\mathrm{final}} \text{ exists.}
\end{cases}
\tag{2}
$$

*Here, $s_T$ is the terminal state, $\sum_j r_j$ is the cumulative reward collected until reaching the subgoal or waypoint, and $t_{\mathrm{wp}}$ is the time step at which $\mathrm{wp}_{\mathrm{final}}$ is reached.*

In FER, *a success transition* records the full cumulative return up to $t'$ and continues the episode, *a stop-on-failure transition* assigns zero return and sets the next state to $s_T$, which immediately truncates the episode so all future returns are forfeited and unreachable or unreliable subgoals are discouraged, and *a partial-success transition* records $\mathrm{wp}_{\mathrm{final}}$ with its accumulated return, localizing how far the attempt progressed and crediting only the reliably executed portion. Together, these signals delineate the reachability frontier, identify unreliable subgoals, and provide consistent high-level targets that reduce unnecessary high-level decisions. This setup separates reliable from unreliable subgoals and suppresses failure-inducing actions, but it can induce conservatism and reduce exploratory coverage because the agent learns to avoid regions associated with failure. To counterbalance this effect, we introduce a decoupled high-level controller comprising a high-level policy $\pi^h$ for exploitation trained on $\mathcal{B}_F^h$ and a complementary exploration policy $\pi^{\mathrm{exp}}$ that promotes coverage. As in prior graph-based HRL, we assume the goal space $\mathcal{G}$ is known for subgoal selection of both $\pi^h$ and $\pi^{\mathrm{exp}}$, and SSE introduces no additional assumptions. Detailed formulations follow.

**High-level Policy $\pi^h$:** In prior work, subgoals are typically selected with Gaussian policies with small noise. Although adequate for those methods, this concentrates exploration near the current maximum subgoal. As noted above, we aim for a high-level policy that can target any point in the goal space, thereby broadening exploration. We therefore define an $\epsilon$-greedy $\pi^h$ as

$$
\pi^h(\tilde{g}_t \mid s_t, g) = \begin{cases}
\tilde{g}_{\mathrm{max},t} := \arg\max_{\tilde{g} \in \mathcal{G}} Q^h(s_t, \tilde{g}, g) & \text{with probability } 1 - \epsilon, \\
\tilde{g}_{\mathrm{rand}} \sim \mathrm{Uniform}(\mathcal{G}) & \text{with probability } \epsilon,
\end{cases}
\tag{3}
$$

where $\tilde{g}_{\mathrm{max},t}$ is the greedy subgoal, in practice generated by the actor network trained to choose the maximum of $Q^h$, $\tilde{g}_{\mathrm{rand}}$ is sampled uniformly from $\mathcal{G}$ to ensure persistent global exploration, and $Q^h$ is trained off-policy using $\mathcal{B}_F^h$. This formulation allows $\pi^h$ to select diverse points in the goal space independently of the agent's current location, which is crucial for broader exploration.

**Exploration Policy $\pi^{\mathrm{exp}}$:** To promote exploration, $\pi^{\mathrm{exp}}$ targets low-density, underexplored regions of the goal space. To do this, we first identify novel regions $V_{\mathrm{novel}} \subset \mathcal{G}$, for which we employ either a grid-based estimator or a model-based approach. In this paper, we consider both: the grid-based estimator is used for simplicity in 2D/3D goal spaces that are widely adopted in real-world settings (e.g., position goal spaces), whereas the model-based approach, although more complex, is adopted to facilitate extension to higher-dimensional goal spaces. The exploration policy $\pi^{\mathrm{exp}}$ is defined as:

$$
\pi^{\mathrm{exp}}(\tilde{g}_t \mid s_t, g) = \begin{cases}
g & \text{with probability } \frac{1}{3}, \\
\tilde{g}_{\mathrm{max},t} & \text{with probability } \frac{1}{3}, \\
\tilde{g}_{\mathrm{novel}} \sim \mathrm{Uniform}(V_{\mathrm{novel}}) & \text{with probability } \frac{1}{3}.
\end{cases}
\tag{4}
$$

where the novel region $V_{\mathrm{novel}}$ is defined in one of two ways, depending on whether we use the grid-based or the model-based learning scheme, that is, $V_{\mathrm{novel}} = V_{\mathrm{novel}}^{\mathrm{grid}}$ or $V_{\mathrm{novel}} = V_{\mathrm{novel}}^{\mathrm{model}}$. The grid-based novel region is defined as $V_{\mathrm{novel}}^{\mathrm{grid}} := C_{\mathcal{G}}^{m_{\mathrm{novel}}}$, where $C_{\mathcal{G}}^m$ denotes the grid cell of size $d_{\mathcal{G}}$ that partitions the goal space $\mathcal{G}$, $m$ indexes the grid cells, and $m_{\mathrm{novel}} = \arg\min_m N(C_{\mathcal{G}}^m)$ is the

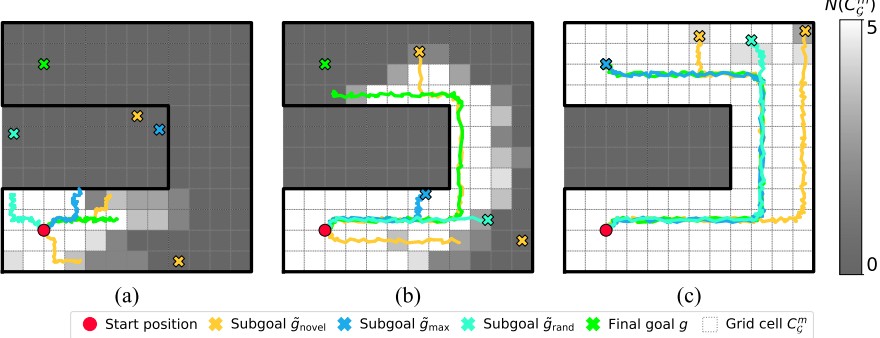

Figure 2: Initial subgoals at $t = 0$ selected by $\pi^h$ and $\pi^{\exp}$, with corresponding Ant agent trajectories at (a) early, (b) intermediate, and (c) final training stages in the U-maze task. The goal space (agent positions in the map) is partitioned into grid cells $C_{\mathcal{G}}^m$. $\pi^h$ selects between $\tilde{g}_{\max}$ and $\tilde{g}_{\text{rand}}$ to encourage broad coverage, while $\pi^{\exp}$ samples from $\tilde{g}_{\text{novel}}$, $\tilde{g}_{\max}$, and $g$ to visit underexplored regions and the goal. Over time, unreachable areas are excluded from subgoal candidates via SSE.

index of the cell with the smallest visitation count $N(C_{\mathcal{G}}^m)$. The model-based novel region is defined as $V_{\text{novel}}^{\text{model}} := \left\{ \arg \max_{v \in V} \| f_\zeta(v) - f_{\zeta_{\text{targ}}}(v) \| \right\}$, where $f_\zeta$ is the learner network with parameter $\zeta$, and $f_{\zeta_{\text{targ}}}$ is the initialized target network proposed by (Burda et al., 2018) to describe novel regions based on prediction error. The learner $f_\theta$ is trained to minimize the prediction error for node samples. Then, the novel subgoal $\tilde{g}_{\text{novel}}$ is uniformly sampled from $V_{\text{novel}}$. Both $\pi^h$ and $\pi^{\exp}$ operate under the same SSE mechanism, where episodes continue only upon successful subgoal completion and terminate on failure. To control exploration, we sample from $\pi^{\exp}$ early in training and then gradually mix it with $\pi^h$ using a ratio $\eta : (1 - \eta)$, where $\eta$ controls exploration strength. This balanced scheme preserves coverage, accelerates the discovery of reachable subgoals, and improves high-level generalization. Implementation details and parameters such as $\eta$ are described in B.2

To illustrate the behavior of the proposed method, Fig. 2 shows grid-wise visitation in the goal space and low-level trajectories toward various initial subgoals at three stages of training: (a) early (10K steps), (b) intermediate (150K steps), and (c) final (500K steps). The environment is a U-maze where a MuJoCo (Todorov et al., 2012) Ant agent navigates to the final goal $g$ located at the upper left of the map. In (a), most of the goal space remains unexplored, and initial subgoals selected by $\pi^{\exp}$ (grid-based) and $\pi^h$, including the final goal, random subgoals, and novel regions, result in wide-ranging trajectories that promote broad exploration. As training progresses in (b), the agent expands its coverage across the map, and in (c), it consistently reaches the goal $g$. Notably, SSE enables the agent to reach any reachable subgoal in a single high-level step, regardless of distance, supporting reliable execution and efficient long-range planning. This allows the agent to solve complex tasks using only a few high-level decisions.

### 4.3 FAILURE-AWARE PATH REFINEMENT

In the SSE framework, reliable subgoal execution is crucial as each subgoal must be reached within a single high-level step. As described in Section 2, graph-based methods use Dijkstra's algorithm to compute waypoint paths from $\phi(s_t)$ to $\tilde{g}_t$ on a graph $G = (V, E)$, with edge distances defined by $d$ in equation 1. However, this distance ignores failure cases like collisions or getting stuck, causing agents to fail even on the shortest path. We observe these failure-prone regions significantly hinder subgoal success. To address this, we introduce a *failure-aware path refinement* strategy that increases edge costs in unreliable regions, steering the planner toward safer alternatives. For each edge from node $v_1$ to node $v_2$, we define the failure ratio $\text{ratio}_{\text{fail}}$ for the target node $v_2$, and refine the edge by increasing its distance as the failure ratio increases as

$$\tilde{d}(v_1 \to v_2) = d(v_1 \to v_2) \times \max\left(1, c_{\text{dist}} \cdot \text{ratio}_{\text{fail}}(v_2)\right), \ \forall v_1, \ v_2 \in V, \tag{5}$$

where $d$ is the original edge distance from equation 1, and $c_{\text{dist}} > 1$ is a scaling factor. For consistency with our exploration policy, the failure ratio for the target node is also defined in two ways, following the grid-based and model-based approaches: $\text{ratio}_{\text{fail}} = \text{ratio}_{\text{fail}}^{\text{grid}}$ or $\text{ratio}_{\text{fail}} = \text{ratio}_{\text{fail}}^{\text{model}}$. The grid-based failure ratio is defined as $\text{ratio}_{\text{fail}}^{\text{grid}}(v_2) := N_{\text{fail}}(C_{\mathcal{G}}^m)/N(C_{\mathcal{G}}^m)$, where $v_2 \in C_{\mathcal{G}}^m$, $N_{\text{fail}}(C_{\mathcal{G}}^m)$ is the number of failure episodes in the cell $C_{\mathcal{G}}^m$. The model-based failure ratio is defined

as $\text{ratio}_{\text{fail}}^{\text{model}}(v_2) := F_\xi(v_2)$, where $F_\xi$ is a failure predictor network with parameters $\xi$ trained over node samples using a cross-entropy objective, assigning label 1 to nodes from failed trajectories and label 0 to nodes from successful trajectories. To ensure sufficient low-level policy competence, the failure-aware path refinement is activated only after $\lambda_{\text{count}}$ successful visits. A higher failure ratio increases the adjusted edge distance, encouraging Dijkstra's algorithm to avoid unreliable regions.

As described above, we instantiate SSE with two variants that differ in their exploration policy and failure-aware path refinement, namely a grid-based approach and a model-based approach. We refer to SSE with the grid-based variant as SSE (Grid) and SSE with the model-based variant as SSE (Model), and we empirically compare these two variants to analyze how the choice of approach affects learning performance.

Fig. 3 illustrates the effect of the proposed grid-based path refinement in a bottleneck environment. In (a), without refinement, the agent repeatedly follows the shortest path through a narrow corridor near a wall, often resulting in failure. In (b), with refinement applied, increased edge costs in high-failure regions steer Dijkstra's algorithm toward safer detours. When no alternatives exist (e.g., in the bottleneck), the agent still passes through, preserving reachability. This demonstrates that the refinement enhances subgoal success while maintaining overall reachability, supporting more stable execution in complex tasks. In summary, the proposed SSE framework is illustrated in Fig. 4, with a condensed version of the algorithm provided in Algorithm 1. The full algorithm and implementation details, including the graph construction and training losses for RL training and model-based approach are provided in Appendix B.

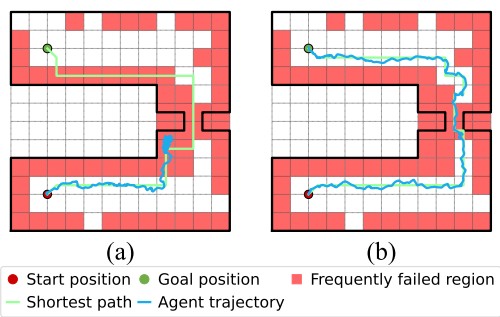

(a)              (b)

● Start position   ● Goal position   ■ Frequently failed region
— Shortest path   — Agent trajectory

Figure 3: Comparison of agent trajectories (blue lines) in a map with a bottleneck: (a) without path refinement and (b) with the proposed path refinement. Green lines represent the shortest waypoint paths computed via Dijkstra's algorithm, while red areas denote grid cells with high failure ratios, i.e., $\text{ratio}_{\text{fail}}(C_{\mathcal{G}}^m) > 0.05$.

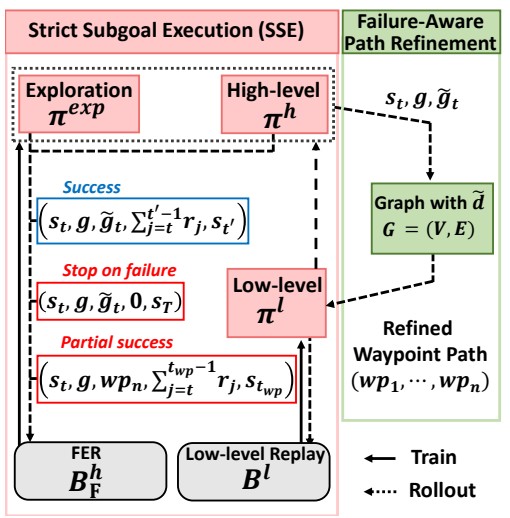

Figure 4: The proposed SSE framework.

**Algorithm 1** Strict Subgoal Execution (SSE)

---

Initialize policies $\pi^h$, $\pi^{\text{exp}}$, $\pi^l$, and graph $G$
**for** *each iteration* **do**
    **for** *each episode* **do**
        **for** *each high-level selection step* **do**
            Sample a subgoal $\tilde{g}_t$ from $\pi^h$ and $\pi^{\text{exp}}$
            Plan a waypoint path from $\phi(s_t)$ to $\tilde{g}_t$
            Roll out low-level policy $\pi^l(s_t, \text{wp}_i) \, \forall i$:
            **if** $\|\phi(s_{t'}) - \tilde{g}_t\| < \lambda$ *(success)* **then**
                Store the success transition and continue
            **else**
                Store the failure and partial-success transitions, then terminate the episode
            **end**
        **end**
    **end**
    Count $N(C_{\mathcal{G}}^m)$ and $N_{\text{fail}}(C_{\mathcal{G}}^m) \, \forall m$ **(Grid-based)**
    Update models $f_\zeta$ and $F_\xi$ **(Model-based)**
    Update $Q^h, Q^l, \pi^h, \pi^l$ via off-policy RL
**end**

---

## 5 EXPERIMENTS

In this section, we evaluate our SSE framework on 9 challenging long-horizon tasks, including 5 AntMaze environments (`U-maze`, `π-maze`, `AntMazeComplex`, `AntMazeBottleneck`,

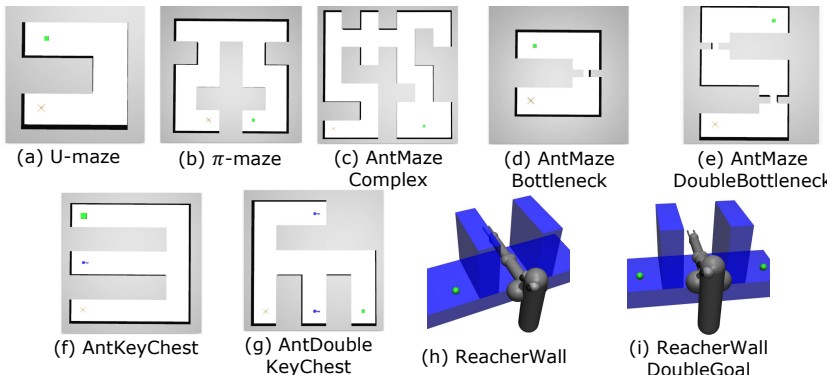

(a) U-maze    (b) π-maze    (c) AntMaze Complex    (d) AntMaze Bottleneck    (e) AntMaze DoubleBottleneck

(f) AntKeyChest    (g) AntDouble KeyChest    (h) ReacherWall    (i) ReacherWall DoubleGoal

Figure 5: Considered long-horizon environments: 5 AntMaze, 2 KeyChest, and 2 Reacher tasks

and `AntMazeDoubleBottleneck`). These range from simple layouts (`U-maze`) to complex structures with narrow corridors (`AntMazeBottleneck`). We also assess 2 KeyChest tasks (`AntKeyChest`, `AntDoubleKeyChest`), where the agent must collect 1 or 2 keys before reaching the final goal, even though the keys are not explicitly defined as goals. Additionally, we evaluate 2 Reacher tasks (`ReacherWall`, `ReacherWallDoubleGoal`), where a 3D robot must navigate obstacles to reach one or two goals. KeyChest and DoubleGoal tasks require intermediate objectives, making them ideal for testing high-level planning. See Fig. 5 for visualizations and Appendix C for details. While the main experiments focus on fixed-goal settings, additional comparisons for random-goal setups are included in Appendix D.1.

## 5.1 PERFORMANCE COMPARISON

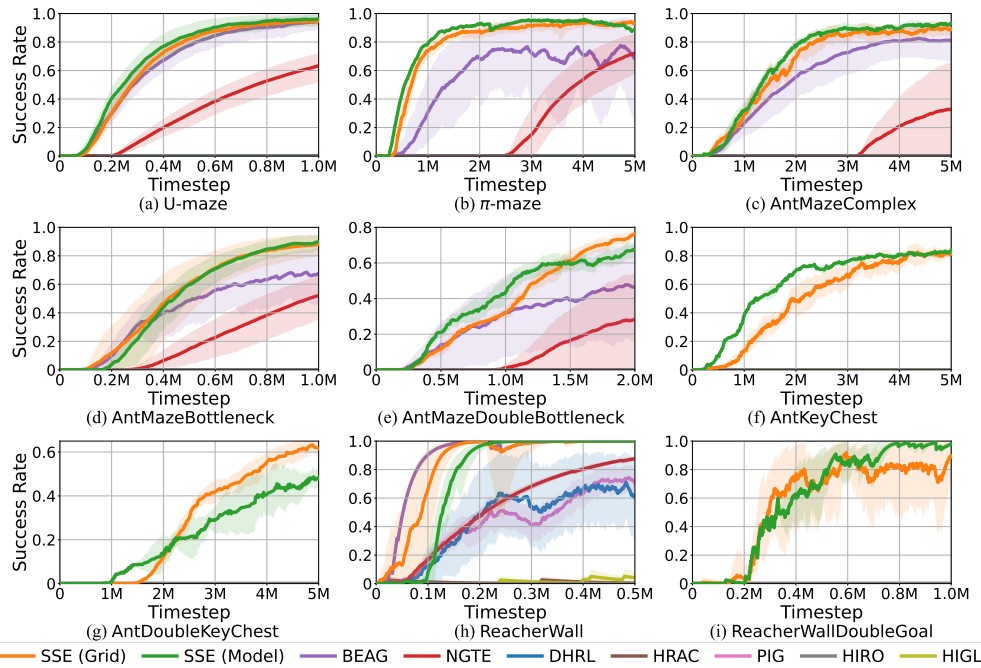

Figure 6: Performance comparison on various long-horizon environments

We compare SSE with a range of hierarchical RL and recent graph-based methods. Specifically, we evaluate 2 HRL approaches: **HIRO** (Nachum et al., 2018b), which improves sample efficiency via hindsight goal relabeling, and **HRAC** (Zhang et al., 2020), which penalizes subgoal selection based on reachability. We also include 3 graph-based HRL methods: **HIGL** (Kim et al., 2021), which applies intrinsic penalties over a graph, **DHRL** (Lee et al., 2022), which finds a path and gives waypoints to low-level as a goal via graph planning, and **NGTE** (Park et al., 2024), which expands

graphs using novelty to better address fixed-goal settings. Additionally, we consider 2 graph-based methods without explicit high-level policies: **PIG** (Kim et al., 2023), which integrates graph structures into imitation learning to skip redundant subgoal actions, and **BEAG** (Yoon et al., 2024), which uses breadth exploration with imaginary landmarks for goal-reaching.

For our proposed framework, we evaluate two variants: a grid-based implementation **SSE (Grid)**, which offers a simple implementation in practical 2D/3D goal spaces, and a learning-based implementation **SSE (Model)**, which is designed to be scalable to higher-dimensional goal spaces, as described in Section 4. Both variants are evaluated with the best hyperparameter settings ($c_{\text{dist}}$, $d_{\mathcal{G}}$, $\eta$) from ablation studies, while all baselines use author-provided implementations. In FER, the threshold $\lambda$ used to determine subgoal reachability is the same as the threshold used to define the low-level reward in Section 2.2. We use the same $\lambda$ values as the baseline methods to ensure a fair and consistent comparison. Detailed descriptions of each algorithm, along with the hyperparameter configurations for our proposed method, are provided in Appendix C.

Fig. 6 shows mean success rates over 5 seeds (solid lines) with standard deviations (shaded). From the results, both SSE variants consistently outperform other graph-based and conventional HRL methods across all benchmarks. Conventional HRL methods (DHRL, HIRO, PIG, HRAC, and HIGL) rely on random goal sampling for exploration but often struggle with fixed-goal tasks, while occasionally succeeding in random-goal setups provided in Appendix D.1, highlighting the increased challenge of fixed goals due to limited exploration opportunities. In relatively simple environments such as `U-maze`, `π-maze`, and `AntMazeComplex`, baseline methods like BEAG, grid-based nodes, and NGTE demonstrate reasonable performance, but SSE typically converges more quickly. In bottleneck environments, SSE further excels by using failure-aware path refinement to avoid unstable regions as shown in Fig. 3. In more complex tasks like KeyChest and `ReacherWallDoubleGoal`, which require reaching intermediate objectives, baseline methods largely fail. Conventional HRL suffers from long high-level horizons, and goal-centric methods without high-level decision-making, such as BEAG, cannot reason about intermediate targets. In contrast, SSE mitigates these issues by strictly enforcing subgoal completion, reducing high-level decision steps. These results highlight the versatility, efficiency, and generalization of the proposed SSE framework. While both variants succeed in all tasks, SSE (Grid) converges faster due to lower overhead. In contrast, SSE (Model) is designed for scalability to high-dimensional spaces where discretization is infeasible, despite higher training complexity. In addition, extended evaluations on `AntPush`, `AntFall`, and `Ant4Rooms` in Appendix E.3 confirm that SSE consistently matches or exceeds baselines. Furthermore, computational analysis in Appendix D.2 demonstrates that SSE achieves lower complexity per iteration than baselines.

## 5.2 FURTHER ANALYSIS AND ABLATION STUDIES

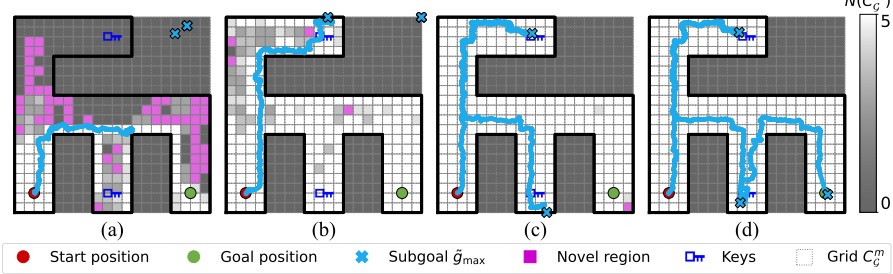

Figure 7: Trajectory analysis for SSE subgoals $\tilde{g}_{\max}$ in `AntDoubleKeyChest` at: (a) early stage, (b) reaches first key, (c) collects both keys, (d) reaches goal after collecting both keys (task success).

In the following analysis and ablation studies, we focus on the grid-based approach and refer to SSE (Grid) simply as SSE, as it consistently achieves better performance and offers a richer set of hyperparameters for analysis and implementation. Fig. 7 presents a trajectory analysis of the proposed SSE framework in the `AntDoubleKeyChest` environment, illustrating how the agent progressively explores the map and collects both keys and the final goal. In the early stage (a) ($t \approx 300K$), the agent expands map coverage by sampling diverse subgoals, similar to simpler environments. As training progresses, the agent visits increasingly more regions, allowing it to reach the first key within a single high-level step, as shown in (b) ($t \approx 1M$). In a more advanced stage (c) ($t \approx 1.5M$), it collects both keys in just two high-level steps. Eventually, as shown in (d) ($t \approx 3M$),

the agent completes the entire task, including both keys and the goal, in only three high-level steps. These results demonstrate that SSE allows the agent to reach any location in the map with a single high-level decision. As a result, it can solve complex multi-goal tasks using a minimal number of high-level steps, which highlights the effectiveness of SSE in long-horizon environments that require sequential decision-making for the high-level policy. Notably, SSE solves this sequential task without an explicit curriculum. The reliability of its high-level policy, learned via FER, combined with an augmented state including key-possession flags, enables the agent to autonomously discover the required sequence of sub-objectives.

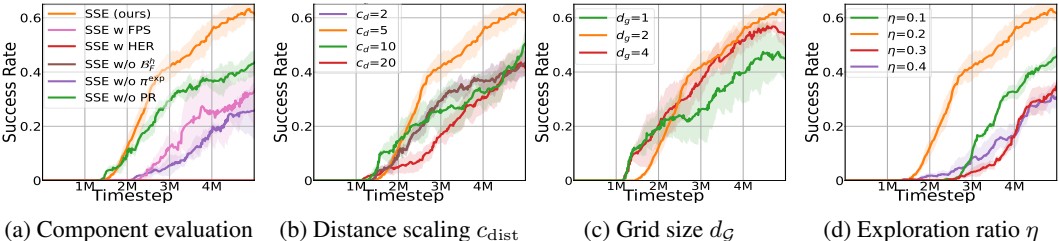

(a) Component evaluation    (b) Distance scaling $c_{\text{dist}}$    (c) Grid size $d_{\mathcal{G}}$    (d) Exploration ratio $\eta$

Figure 8: Ablation study on `AntDoubleKeyChest` environment

We conduct an ablation study on `AntDoubleKeyChest` to evaluate the contribution of each component in SSE and to analyze the impact of key hyperparameters, as shown in Fig. 8. For the component analysis, we consider five variants: (1) **SSE with FPS**, which replaces the grid-based landmarks with Farthest Point Sampling Kim et al. (2021); (2) **SSE with HER**, which replaces strict subgoal execution with a conventional high-level policy that uses HER; (3) **SSE without $\mathcal{B}_F^h$**, which disables FER by using a standard replay buffer; (4) **SSE without $\pi^{\text{exp}}$**, which disables the exploration policy; and (5) **SSE without path refinement (PR)**, which disables the failure-aware adjustment. All variants exhibit degraded performance. Notably, the settings using HER or disabling FER fail to learn entirely, emphasizing the importance of strict subgoal execution and the consistent high-level signals provided by FER. SSE with FPS eventually succeeds, though less efficiently than the grid version, while still outperforming FPS-based baselines. This confirms that our framework remains effective regardless of the specific landmark selection strategy. Our hyperparameter analysis shows that setting $c_{\text{dist}} = 5$ achieves an optimal balance, though performance holds steady across a range of values. Similarly, performance also holds steady for grid resolution, though overly fine grids (e.g., $d_{\mathcal{G}} = 1$) can slow learning. For the exploration ratio $\eta$, a value of 0.2 proves optimal. While hyperparameter tuning optimizes performance, SSE consistently outperforms all baselines. Additional analyses in other environments and sensitivity studies on the reachability threshold $\lambda$ and the high-level discount factor $\gamma^h$ are provided in Appendix E, further characterizing SSE.

## 6 LIMITATION

Our framework introduces new hyperparameters, such as the exploration ratio $\eta$ and path refinement factor $c_{\text{dist}}$, which require tuning. However, we find their effective ranges to be stable across diverse environments, and our ablation studies confirm that the framework maintains strong performance across a wide range of values, minimizing the overall tuning cost. While SSE also adds computational steps, its principle of early termination on subgoal failure yields significant efficiency gains. This prevents long, unproductive trajectories and results in a faster computation time compared to other recent methods, as quantitatively analyzed in Appendix D.2.

## 7 CONCLUSION

In this paper, we proposed SSE, a graph-based HRL framework designed to improve reliability and efficiency in long-horizon, goal-conditioned tasks. By enforcing single-step subgoal reachability, SSE enables more direct and reliable high-level planning and significantly reduces decision horizons. The introduction of failure-aware path refinement and a decoupled exploration policy further enhances subgoal reliability and map coverage. Extensive experiments demonstrate that SSE consistently outperforms existing HRL and graph-based methods across a range of complex tasks. These results highlight the framework's effectiveness in enabling stable and generalizable behavior, making it a promising approach for scalable hierarchical control in various long-horizon tasks.

## ACKNOWLEDGMENT

This work was supported partly by the Institute of Information & Communications Technology Planning & Evaluation (IITP) grant funded by the Korea government (MSIT) (No. RS-2022-II220469, Development of Core Technologies for Task-oriented Reinforcement Learning for Commercialization of Autonomous Drones), (No. RS-2025-25442824, AI Star-Fellowship Program (UNIST)), and (No. RS-2020-II201336, Artificial Intelligence Graduate School Support (UNIST)), and partly by the National Research Foundation of Korea (NRF) grant funded by the Korea government (MSIT) (No. RS-2025-23523191, LLM-Based Multi-Agent Reinforcement Learning for End-to-End Large Autonomous Swarm Control).

## ETHICS STATEMENT

This paper introduces a foundational algorithm, SSE, designed to improve the long-horizon planning capabilities of reinforcement learning agents. All experiments were conducted in standard, simulated robotics environments. As such, our research does not involve human subjects, sensitive or personally identifiable data, nor does it directly address systems that interact with people. Therefore, issues of data privacy, dataset bias, and fairness are not directly applicable to this work.

## REPRODUCIBILITY STATEMENT

We are committed to the reproducibility of our research. The complete source code for our proposed framework, SSE, and all experiments is included as an anonymized zip file in the supplementary materials. A detailed breakdown of the implementation, including network architectures and training procedures, can be found in Appendix B. All hyperparameters required to reproduce our results are provided in Appendix C.3, with common settings listed in Table 2 and environment-specific configurations in Table 3. Furthermore, Appendix C details the full experimental setup, including descriptions of the baseline algorithms, specifications for all environments (Table 1), and the hardware and software configurations on which the experiments were conducted. We believe these resources will enable the reproduction of our findings.

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

## A  THE USE OF LARGE LANGUAGE MODELS

We utilized a large language model (LLM) as an assistive tool during the preparation of this manuscript. The LLM's role was strictly limited to polishing the text, which includes improving clarity, conciseness, and correcting grammatical errors. The LLM was not used for research ideation. The authors have carefully reviewed and edited all content and take full responsibility for the scientific accuracy and integrity of this work. The LLM is not credited as an author.

## B  IMPLEMENTATION DETAILS

This section presents additional implementation details of the proposed SSE framework. The graph construction method for the proposed SSE is detailed in Appendix B.1, while the training dynamics and implementation specifics of the framework are described in Appendix B.2. Each subsection highlights the design motivations and practical considerations for each module or mechanism.

### B.1  GRAPH CONSTRUCTION OF SSE

In this section, we describe how existing methods define the graph $G = (V, E)$, which consists of a landmark node set $V$ and the edge set $E$ with edge distances $d$. We then explain how the proposed SSE framework constructs this graph.

**Landmark Node Set $V$ Construction**

Existing HRL methods such as DHRL (Lee et al., 2022) and NGTE (Park et al., 2024) construct the landmark node set $V$ by selecting visited states during exploration. They employ the Farthest Point Sampling (FPS) algorithm to identify landmark nodes that are far apart from each other. This approach ensures that frequently visited regions are well-represented, as it builds $V$ based on actual agent trajectories. However, it is limited to visited states, meaning unexplored or infrequently visited areas cannot be selected as landmarks, slowing exploration in those regions. In contrast, BEAG (Yoon et al., 2024) accelerates exploration by partitioning the goal space into a grid structure, creating landmark nodes at each grid intersection. This method allows for faster exploration by using virtual goal positions as landmarks, independent of visitation frequency. The structured grid layout enables more systematic and efficient exploration. To leverage this advantage and ensure a structured, reproducible setup, SSE adopts a grid-based landmark selection strategy inspired by BEAG. Given a 2D goal space of size $x \times y$ and a grid size of $d_{\mathcal{G}}$, the landmark set $V$ is constructed as follows:

$$V = \{(i \cdot d_{\mathcal{G}}, j \cdot d_{\mathcal{G}}) \in \mathcal{G} \mid i = 0, \cdots, \frac{x}{d_{\mathcal{G}}} - 1, \ j = 0, \cdots, \frac{y}{d_{\mathcal{G}}} - 1\}. \tag{6}$$

For a 3D goal space, the landmark node set is defined as $V = \{(i \cdot d_{\mathcal{G}}, j \cdot d_{\mathcal{G}}, k \cdot d_{\mathcal{G}}) \in \mathcal{G} \mid i = 0, \cdots, \frac{x}{d_{\mathcal{G}}} - 1, \ j = 0, \cdots, \frac{y}{d_{\mathcal{G}}} - 1, \ k = 0, \cdots, \frac{z}{d_{\mathcal{G}}} - 1\}$. By constructing landmark nodes in this grid-based manner, SSE achieves faster and more structured exploration compared to visitation-based methods, ensuring efficient path planning and reliable subgoal execution.

**Definition of Edge Distance**

Given the landmark node set $V$, the edge set $E$ is defined as the collection of distances $d(v_1 \rightarrow v_2)$ between any two nodes $v_1, v_2 \in V$. For previous graph-based RL methods, the edge distance $d$ is computed as $d(v_1 \rightarrow v_2) := \log_{\gamma^l} \left(1 + (1 - \gamma^l)Q^G(v_1, v_2, \pi^l)\right)$, where $\gamma^l$ is the low-level discount factor used for training the low-level policy $\pi^l$, and $Q^G$ is the value function estimating the traversal cost from $v_1$ to $v_2$, as described in Section 2. Existing HRL methods like DHRL (Lee et al., 2022) and NGTE (Park et al., 2024) directly use the low-level value function $Q^l$, which is trained with a step-based reward of $-1$, to define the distance as the expected number of steps required for the low-level policy to navigate from $v_1$ to $v_2$. Although this method reflects actual navigation costs, it is sensitive to instability during $Q^l$ training, resulting in fluctuating edge distances. In contrast, BEAG (Yoon et al., 2024) measures the distance using the Euclidean norm $d_E = ||v_1 - v_2||$, providing a stable but less accurate representation of traversal costs. To leverage the strengths of both approaches, SSE defines the edge distance $d(v_1 \rightarrow v_2)$ by combining $Q^l$ and $d_E$ as follows:

$$d(v_1 \rightarrow v_2) = \frac{1}{2} \left[\log_{\gamma^l} \left(1 + (1 - \gamma^l)Q^l(v_1, v_2, \pi^l)\right) + \log_{\gamma^l} \left(1 + (1 - \gamma^l)d_E(v_1, v_2)\right)\right]. \tag{7}$$

This hybrid formulation allows SSE to benefit from the stability of Euclidean distances when $Q^l$ is not fully converged, while still capturing the true traversal cost as $Q^l$ improves. As a result, the edge set $E$ is constructed as $E = \{d(v_1 \rightarrow v_2) \mid v_1, v_2 \in V\}$. Here, $d$ represents the raw edge distance before failure-aware path refinement is applied, ensuring both stability and adaptive accuracy in path estimation.

## B.2 Detailed Loss Functions and Implementation of the SSE Framework

**RL training losses** As described in Section 4, the proposed SSE framework is an HRL structure that employs a high-level policy $\pi^h$, an exploration policy $\pi^{\mathrm{exp}}$, and a low-level policy $\pi^l$. The exploration policy does not require separate parameterization for high-level actions, whereas $\pi^h$ and $\pi^l$ are parameterized by $\theta^h$ and $\theta^l$, respectively, and are represented as $\pi^h_{\theta^h}$ and $\pi^l_{\theta^l}$. To evaluate these policies, SSE defines the parameterized high-level value function $Q^h_{\psi^h}$ and the low-level value function $Q^l_{\psi^l}$, where $\psi^h$ and $\psi^l$ are the respective parameters. As mentioned in Section 2, the high-level policy and value function are trained to maximize external rewards, while the low-level policy and value function are optimized to reach designated waypoints incrementally. To facilitate this, the low-level policy receives the reward $r^l_t$ at each time step, defined as follows:

$$r^l_t = \begin{cases} 0 & \text{if } \|\phi(s_{t+1}) - \mathrm{wp}_i\| < \lambda, \text{ (agent reaches the current waypoint)} \\ -1 & \text{otherwise,} \end{cases} \tag{8}$$

where $\lambda$ is the reachability check threshold, $\mathrm{wp}_i$ is the target waypoint at the current timestep $t$, and $(\mathrm{wp}_1, \cdots, \mathrm{wp}_n)$ represent the shortest path from $\phi(s_t)$ to the subgoal $\tilde{g}_t$. The high-level and low-level policies, along with their value functions, are trained using the transitions stored in the FER $\mathcal{B}^h_F$ and the low-level buffer $\mathcal{B}^l$ through the TD3 algorithm (Fujimoto et al., 2018), a standard off-policy RL method. The value function losses for high-level and low-level policies are defined as follows:

$$\mathcal{L}_{Q^h}(\psi^h) = \mathbb{E}_{B^h_F} \left[ \left( Q^h(s_t, \tilde{g}_t) - \left( r^h_t + \gamma^h \min_{i=1,2} Q^h_{\bar{\psi}^h_i}(s_{t'}, \pi^h_{\theta^h}(s_{t'}, g)) \right) \right)^2 \right]$$

$$\mathcal{L}_{Q^l}(\psi^l) = \mathbb{E}_{B^l} \left[ \left( Q^l(s_t, a_t) - \left( r^l_t + \gamma^l \min_{i=1,2} Q^l_{\bar{\psi}^l_i}(s_{t+1}, \pi^l_{\theta^l}(s_{t+1}, \mathrm{wp}_i)) \right) \right)^2 \right], \tag{9}$$

where $t'$ denotes the termination time of the low-level path execution, which is variable as the step concludes only upon success or failure. In the case of a partial success, $t_{\mathrm{wp}}$ (used to store transitions in FER) indicates the time step when the agent reached the last successful waypoint $wp_{\mathrm{final}}$. The high-level reward is then defined as $r^h_t = \sum_{j=t}^{t'-1} r_j$ for a successful trajectory and $r^h_t = 0$ for a failed one. The terms $\bar{\psi}^h$ and $\bar{\psi}^l$ are target network parameters updated via exponential moving average (EMA), and $\gamma^h$ and $\gamma^l$ are the discount factors for training the high-level and low-level policies, respectively. The actor losses for optimizing the policies are defined as follows:

$$\mathcal{L}_{\pi^h}(\theta^h) = -\mathbb{E}_{B^h_F} \left[ Q^h_{\psi^h}(s_t, \pi^h_{\theta^h}(s_t, g)) \right], \quad \mathcal{L}_{\pi^l}(\theta^l) = -\mathbb{E}_{B^l} \left[ Q^l_{\psi^l}(s_t, \pi^l_{\theta^l}(s_t, \mathrm{wp}_i)) \right]. \tag{10}$$

The parameters are optimized using the Adam optimizer (Kingma, 2014) to minimize the respective loss functions. SSE distinguishes between the high-level discount factor $\gamma^h$ and the low-level discount factor $\gamma^l$, setting $\gamma^l = 0.99$ as typical in RL, and $\gamma^h = 0.4$ to limit return propagation across high-level steps, encouraging shorter path optimization. In the initial stages of training, rewards are sparse. To address this, the FER $\mathcal{B}^h_F$ is divided evenly, with half storing successful trajectories and the other half storing trajectories with zero reward. This design improves learning signals from successful experiences. The exploration ratio, which controls the balance between the exploration policy and the high-level policy, starts at 1:0 and decays by 0.05 per iteration until it reaches $\eta{:}(1-\eta)$, as outlined in Section 4. This scheduling promotes exploration initially and shifts the focus to high-level learning as training progresses. If the low-level agent fails to reach the high-level subgoal $\tilde{g}_t$, the trajectory is marked as failed. In cases where the agent becomes stuck, such as flipping over or hitting obstacles, its position may remain unchanged for long periods. To improve sample efficiency, if no movement is detected for 500 steps, the trajectory is classified as failed and the episode is terminated. The complete SSE algorithm is provided in Algorithm 2.

**Model training losses for SSE (Model)**   We now describe the implementation details of the prediction network $f_\zeta$, which computes the prediction error used to determine the novel region in the model-based approach, and the failure predictor network $F_\xi$, which estimates the failure ratio of target nodes.

In the model-based variant of the exploration policy, we estimate the novel region $V_{\text{novel}}^{\text{model}}$ using Random Network Distillation (RND) (Burda et al., 2018), which was originally proposed in standard RL to identify novel states based on prediction error. Following the RND formulation, we use a fixed target network $f_{\zeta_{\text{targ}}} : V \to \mathbb{R}^k$ and a predictor network $f_\zeta : V \to \mathbb{R}^k$ with parameters $\zeta$. The predictor $f_\zeta$ is trained on visited sample nodes stored in the buffer $\mathcal{D}$ to regress the target output via the mean squared error objective

$$L_{\text{RND}}(\zeta) = \mathbb{E}_{v \sim \mathcal{D}_{\text{visited}}} \left[ \left\| f_\zeta(v) - f_{\zeta_{\text{targ}}}(v) \right\|_2^2 \right]. \tag{11}$$

As the predictor network is trained to minimize the prediction error on visited nodes, the prediction error becomes small in frequently explored regions and remains large for underexplored nodes. We therefore use the squared prediction error $\left\| f_\zeta(v) - f_{\zeta_{\text{targ}}}(v) \right\|_2^2$ as a novelty score, and nodes with larger error are treated as more novel. By selecting nodes with high prediction error as exploration targets, the model-based variant is encouraged to visit underexplored regions of the goal space.

For failure-aware refinement, we train a failure predictor network $F_\xi : V \to [0, 1]$ parameterized by $\xi$, which outputs the estimated failure probability of each node. The network is trained using node samples stored in buffer $\mathcal{D}_v$, where each node $v$ is paired with a binary label $y \in \{0, 1\}$ indicating whether the trajectory containing $v$ failed ($y = 1$) or succeeded ($y = 0$). We optimize $F_\xi$ with the binary cross-entropy loss

$$L_F(\xi) = -\mathbb{E}_{(v,y) \sim \mathcal{D}_v} \left[ y \log(F_\xi(v)) + (1 - y) \log(1 - F_\xi(v)) \right]. \tag{12}$$

Through this training, $F_\xi(v)$ approximates the empirical failure ratio at node $v$, which we use as the model-based failure estimate in our refinement scheme.

---

**Algorithm 2** Strict Subgoal Execution (SSE)

---

**Input:** Graph $G = (V, E)$, goal $g$, mapping function $\phi$, threshold $\lambda$, exploration ratio $\eta$
**Initialize:** Policies $\pi^h$, $\pi^{\text{exp}}$, $\pi^l$, buffers $B_F^h$, $B^l$, grid cells $C_{\mathcal{G}}$ (Grid) or Models $f_\zeta$, $F_\xi$ (Model)
**for** each iteration **do**
  **for** each episode **do**
    Select the behavior policy: $\pi \leftarrow \pi^{\text{exp}}$ with probability $\eta$, otherwise $\pi \leftarrow \pi^h$;
    **for** each high-level selection step **do**
      Sample a subgoal $\tilde{g}_t \sim \pi(s_t, g)$
      Plan the waypoint path $\text{wp}_{1:n}$ from $\phi(s_t)$ to $\tilde{g}_t$ using Dijkstra's algorithm over $G$ with $\tilde{d}$
      **for** $i = 1, \cdots, n$ **do**
        Roll out the low-level policy $\pi^l(s_t, \text{wp}_i)$ to reach the waypoint $\text{wp}_i$
        Store the $t' - t$ transitions $(s_t, \text{wp}_i, a_t, r_t, s_{t+1})$ into the low-level buffer $\mathcal{B}^l$
        Compute the reward sum: $r_t^h = \sum_{j=t}^{t'-1} r_j$
        **Construct the FER:**
        **if** $\|\phi(s_{t+1}) - \tilde{g}_t\| < \lambda$ **then**
          **Success**: Store the success transition $(s, g, \tilde{g}_t, r_{\text{sum}}, s_{t'})$ into FER $B_F^h$
        **else**
          **Failure**: Store the stop-on-failure transition $(s, g, \tilde{g}_t, 0, s_T)$ into FER $B_F^h$
          Store the partial success transition $(s, g, \text{wp}_{\text{final}}, \sum_{j=t}^{t_{\text{wp}}-1} r_j, s_{t_{\text{wp}}})$ into FER $B_F^h$
          Terminate the episode
        **end**
      **end**
    **end**
  **end**
  Update total visitation count $N(C_{\mathcal{G}}^m)$ and failure count $N_{\text{fail}}(C_{\mathcal{G}}^m)$ for all $C_{\mathcal{G}}^m$ **(Grid-based)**
  Update model parameters $\zeta, \xi$ to minimize $L_{\text{RND}}(\zeta)$ and $L_F(\xi)$ **(Model-based)**
  Update $\psi^h, \theta^h$ using samples from FER $B_F^h$ to minimize $\mathcal{L}_{Q^h}(\psi^h), \mathcal{L}_{\pi^h}(\theta^h)$
  Update $\psi^l, \theta^l$ using samples from $\mathcal{B}^l$ to minimize $\mathcal{L}_{Q^l}(\psi^l), \mathcal{L}_{\pi^l}(\theta^l)$
**end**

---

## C   EXPERIMENTAL SETUP

Our proposed framework is designed to be modular and general, enabling integration with a wide range of baseline methods. For comparison, we employ the official codebases provided by the original authors for HIRO, HRAC, HIGL, DHRL, NGTE, and BEAG. All baselines are run using the hyperparameters specified in their respective publications, and conducted on an NVIDIA RTX 3090 GPU with an Intel Xeon Gold 6348 CPU (Ubuntu 20.04). Appendix C.1 provides descriptions of the baseline algorithms along with links to their official code repositories. Appendix C.2 provides the specifications for the nine environments shown in Fig.9, including their action and observation spaces, goal configurations, and episode horizons. The SSE-specific hyperparameter configurations for each environment are summarized in AppendixC.3.

### C.1   DETAILS OF OTHER BASELINES

- **HIRO** (Nachum et al., 2018b) introduces a hierarchical architecture with relabeling of high-level transitions to account for changing low-level policies, thereby improving off-policy sample efficiency and stability. Open-source code of HIRO is available at https://github.com/watakandai/hiro_pytorch

- **HRAC** (Zhang et al., 2020) adds a learned adjacency constraint to ensure subgoal feasibility. It penalizes high-level selections that attempt transitions deemed unreachable within a limited horizon, thereby guiding the agent to learn feasible subgoal structures. Open-source code of HRAC is available at https://github.com/trzhang0116/HRAC

- **HIGL** (Kim et al., 2021) incorporates a coverage-driven and novelty-driven landmark selection strategy. It performs graph-based planning via shortest paths and uses adjacency rewards to guide learning toward under-explored regions. Open-source code of HIGL is available at https://github.com/junsu-kim97/HIGL

- **DHRL** (Lee et al., 2022) constructs a goal graph using Farthest Point Sampling and learns high-level behavior by planning over the graph. It emphasizes temporal abstraction and long-horizon planning via graph traversal and Q-learning. Open-source code of DHRL is available at https://github.com/jayLEE0301/dhrl_official

- **BEAG** (Yoon et al., 2024) employs value-function-driven imaginary landmarks to facilitate exploration of unvisited areas. It estimates landmark distances from a learned value function without relying solely on previously visited states, enabling efficient generalization. Open-source code of BEAG is available at https://github.com/ml-postech/BEAG

- **NGTE** (Park et al., 2024) drives novelty-based exploration by identifying frontier nodes (outposts) and prioritizing expansion toward less-visited regions of the goal graph, encouraging broad and diverse exploration. Open-source code of NGTE is available at https://github.com/ihatebroccoli/NGTE

### C.2   ENVIRONMENTAL DETAILS

We follow standard benchmarks and configurations widely adopted in prior hierarchical reinforcement learning studies (Lee et al., 2022; Park et al., 2024; Yoon et al., 2024), and introduce several new environments designed to evaluate high-level decision-making capabilities such as `AntKeyChest` or `AntDoubleKeyChest`. The environments used in our experiments are visualized in Fig. 9, and their characteristics are described in Table 1.

`AntKeyChest` and `AntDoubleKeyChest` include key flags that are toggled from 0 to 1 when the agent successfully reaches the corresponding key location under a predefined condition. `ReacherWallDoubleGoal` contains two distinct goal positions, and two goal vectors are provided along with corresponding success flags, which are set to 1 upon reaching each goal. The success threshold for determining whether a target is reached is 5 in the AntMaze environments and 0.25 in the Reacher environments.

### C.3   HYPERPARAMETER SETUP

Table 2 summarizes the common hyperparameters used across all environments. These configurations are based on a combination of parameter tuning and default settings from baseline implementations. In particular, the buffer size, batch size, and network architecture follow the original baseline

Table 1: Summary of environment specifications.

| Environment | Spaces (Obs / Action) | Goal space | Reward | Start / Goal Position | Episode length |
|---|---|---|---|---|---|
| U-Maze | Obs: 29-Dof Action: 8-Dof | [-4,20]× [-4,20] | 1 if goal reached else 0 | Start: (0,0) Goal: (0,16) | 600 |
| $\pi$-Maze | Obs: 29-Dof Action: 8-Dof | [-4,36]× [-4,36] | 1 if goal reached else 0 | Start: (8,0) Goal: (24,0) | 1000 |
| AntMazeComplex | Obs: 29-Dof Action: 8-Dof | [-4,52]× [-4,52] | 1 if goal reached else 0 | Start: (0,0) Goal: (40,0) | 2000 |
| AntMazeBottle-neck | Obs: 29-Dof Action: 8-Dof | [-4,20]× [-4,20] | 1 if goal reached else 0 | Start: (0,0) Goal: (0,16) | 600 |
| AntMazeDouble-Bottleneck | Obs: 29-Dof Action: 8-Dof | [-4,20]× [-4,36] | 1 if goal reached else 0 | Start: (0,0) Goal: (16,32) | 1200 |
| AntKeyChest | Obs: 30-Dof Action: 8-Dof | [-4,36]× [-4,36] | 1 if key reached 5 if goal reached with key | Start: (0,0) Key: (0,16) Goal: (0,32) | 2000 |
| AntDouble-KeyChest | Obs: 31-Dof Action: 8-Dof | [-4,36]× [-4,36] | 1 if key1 reached 1 if key2 reached with key1 5 if goal reached with two keys | Start: (0,0) Key1: (16, 32) Key2: (16, 0) Goal: (32, 0) | 3000 |
| ReacherWall | Obs: 17-Dof Action: 7-Dof | [-1,1]× [-1,1]× [-1,1] | 1 if goal reached | Start: (0.99, −0.19, 0) Goal: (0.6, 0.6, −0.1) | 100 |
| ReacherWall-DoubleGoal | Obs: 22-Dof Action: 7-Dof | [-1,1]× [-1,1]× [-1,1] | 1 if goal reached 5 if both goals reached | Start: (0.99, −0.19, 0) Goal1: (0.4, 0.4, −0.1) Goal2: (0.4, −0.8, −0.1) | 200 |

(a) U-maze  (b) $\pi$-maze  (c) AntMaze Complex  (d) AntMaze Bottleneck  (e) AntMaze DoubleBottleneck

(f) AntKeyChest  (g) AntDouble KeyChest  (h) ReacherWall  (i) ReacherWall DoubleGoal

Figure 9: Considered long-horizon environments: 5 AntMaze, 2 KeyChest, and 2 Reacher tasks

code. Learning rates for each policy level, as well as the discount factors, were determined through a structured parameter search.

Table 3 presents environment-specific hyperparameters, including the minimum epsilon threshold $\epsilon_{\min}$, the path refinement scaling factor $c_{\text{dist}}$, the reachability check threshold $\lambda$, and the grid resolution $d_{\mathcal{G}}$. As mentioned in Section 2.2, the threshold $\lambda$ is the standard, environment-provided value used to check for low-level reachability (see equation 8), which is also used consistently across baselines. These values were selected based on targeted parameter searches conducted for each environment.

In the case of `AntDoubleKeyChest`, a higher value of $\epsilon_{\min}$ is required compared to other environments. This is because the task involves discovering multiple intermediate objectives, which increases the need for broad exploration. Other parameters such as the exploration ratio $\eta$ and the refinement scaling factor $c_{\text{dist}}$ were also selected through environment-specific tuning, with further analyses reported in Appendix E.2.

Table 2: Common hyperparameter settings used in SSE.

| Hyperparameter | Value |
|---|---|
| Optimizer | Adam |
| Replay buffer size | 2,500,000 |
| Batch size | 1024 |
| High-level actor learning rate $\alpha^h_{\text{actor}}$ | 0.000005 |
| High-level critic learning rate $\alpha^h_{\text{critic}}$ | 0.00005 |
| Low-level actor learning rate $\alpha^l_{\text{actor}}$ | 0.0001 |
| Low-level critic learning rate $\alpha^l_{\text{critic}}$ | 0.001 |
| High-level discount factor $\gamma^h$ | 0.4 |
| Low-level discount factor $\gamma^l$ | 0.99 |
| Target update rate $\tau$ | 0.005 |

Table 3: Scenario-specific hyperparameters for SSE.

| Scenario | $\epsilon_{\min}$ | $c_{\text{dist}}$ | $\lambda$ | $d_{\mathcal{G}}$ | $\eta$ |
|---|---|---|---|---|---|
| **AntMaze** | | | | | |
| U-maze | 0.1 | 5.0 | 0.5 | 2 | 0.1 |
| AntMazeBottleneck | 0.1 | 5.0 | 0.5 | 2 | 0.1 |
| $\pi$-maze | 0.1 | 5.0 | 0.5 | 2 | 0.2 |
| AntMazeComplex | 0.1 | 5.0 | 0.5 | 2 | 0.2 |
| AntMazeDoubleBottleneck | 0.1 | 10.0 | 0.5 | 2 | 0.1 |
| AntKeyChest | 0.1 | 5.0 | 0.5 | 2 | 0.2 |
| AntDoubleKeyChest | 0.2 | 5.0 | 0.5 | 2 | 0.2 |
| **Reacher** | | | | | |
| ReacherWall | 0.1 | 5.0 | 0.25 | 1 | 0.2 |
| ReacherWallDoubleGoal | 0.1 | 5.0 | 0.25 | 1 | 0.2 |

# D    ADDITIONAL COMPARATIVE EXPERIMENTS

This section presents additional comparative experiments to further assess the generality and computational efficiency of the proposed SSE framework. We evaluate performance in a random-goal training setup in Appendix D.1, where goals are sampled uniformly from the entire reachable state space. Appendix D.2 evaluates computational characteristics, including convergence speed and per-episode compute time.

## D.1    PERFORMANCE COMPARISON UNDER RANDOM GOAL SETUPS

Figure 10 presents experimental results under a random goal setting where goals are sampled uniformly from the entire valid state space during training. This setup contrasts with the fixed-goal

scenarios discussed in the main text. The reward signal remains sparse and poses a significant challenge for goal discovery and policy optimization. Graph-based methods such as BEAG, NGTE, and SSE maintain strong performance even under random goal sampling. These methods autonomously expand their subgoal graph during exploration and achieve success rates comparable to those in the fixed-goal setting. DHRL also benefits from the randomized goal distribution, particularly in simpler environments like `U-maze` and `AntMazeBottleneck` where the agent encounters training signals more frequently. Conversely, methods such as HIRO, HIGL, and HRAC exhibit limited progress in most environments under random-goal conditions. This performance drop stems from their reliance on fixed-frequency subgoal selection and the lack of structured exploration mechanisms suitable for sparse reward settings. In environments requiring multi-stage reasoning such as `AntKeyChest`, SSE is the only method that consistently discovers the key and solves the full task. While NGTE is hierarchical and exploration-driven, it only occasionally learns to acquire the key and exhibits a very low overall success rate under this configuration.

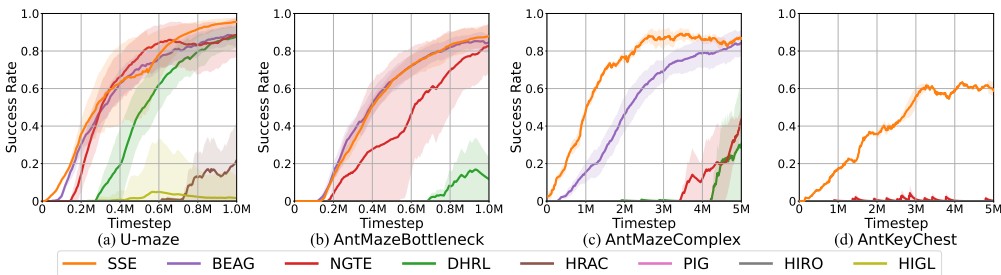

Figure 10: Performance comparison in random goal setting

## D.2 COMPUTATIONAL COMPLEXITY COMPARISON

Table 4 reports the average per-episode computation time and the number of episodes required to reach a 60% success rate. We compare SSE against NGTE and BEAG as they are the only baselines that succeed in at least one of the considered fixed-goal settings. This makes them the most relevant references for evaluating performance and sample efficiency in sparse reward environments. SSE achieves faster convergence across tasks due to its early termination of episodes upon subgoal failure and a reduced number of transitions per episode. These features minimize unnecessary computation and accelerate learning.

This advantage is particularly evident in long-horizon tasks such as `AntKeyChest`. In these tasks, conventional HRL methods consume many timesteps even during failed attempts and require extended horizons for high-level planning. In shorter-horizon scenarios like `AntMazeBottleneck`, the computational benefits of SSE are less significant since subgoal transitions and failure terminations occur less frequently. The operations introduced by SSE, including the evaluation of subgoal completion and uniform sampling, are lightweight and have linear time complexity. Consequently, SSE imposes minimal computational overhead while maintaining stable training dynamics.

Table 4: Average per-episode computation time (in seconds) and the number of episodes required to reach 60% success rate.

| Scenario | SSE | BEAG | NGTE |
|---|---|---|---|
| `AntMazeBottleneck` | 5.94s
432 episodes | 6.46s
425.2 episodes | 12.65s
725.6 episodes |
| `AntKeyChest` | 18.13s
1658 episodes | 57.29s
fail | 119.07s
fail |

## E  EXTENDED ANALYSES OF THE PROPOSED SSE FRAMEWORK

To further validate the design and generality of SSE, this section presents extended analyses across several dimensions. Appendix E.1 provides qualitative trajectory visualizations that illustrate how

SSE dynamically adapts its subgoal planning over time. Appendix E.2 presents ablation and sensitivity analyses to isolate the contribution of individual components and hyperparameters, including the path refinement scale $c_{\text{dist}}$, grid resolution $d_{\mathcal{G}}$, and exploration ratio $\eta$.

### E.1 TRAJECTORY ANALYSIS IN OTHER ENVIRONMENTS

To gain deeper insight into the behavior of SSE, we present trajectory analyses across representative environments.

Fig. 11 illustrates learning progression in `AntMazeBottleneck`. In the early stage (a), the agent has not yet discovered feasible subgoals, causing the high-level policy $\pi^h$ to select subgoals largely at random. During this phase, the exploration policy promotes coverage by gradually expanding into novel regions. In the mid stage (b), the agent fails to traverse the narrow bottleneck, primarily because the low-level policy $\pi^l$ lacks sufficient experience in that region and the associated failure statistics $\rho_{\text{f}}$ remain underrepresented. By the final stage (c), SSE successfully leverages failure-aware refinement and subgoal selection. This allows $\pi^h$ to navigate through the bottleneck and reach the goal reliably.

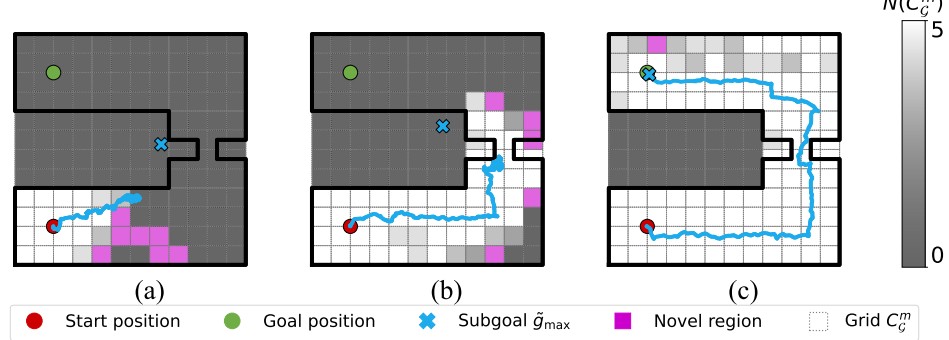

Figure 11: Trajectory analysis of SSE subgoals $\tilde{g}_{\max}$ in `AntMazeBottleneck` across: (a) early stage, (b) mid training, and (c) task success.

A similar trend is observed in `AntMazeDoubleBottleneck`, as shown in Fig. 12. Initially (a), most regions are unexplored, and subgoal selection remains uninformed. In the intermediate phase (b), the agent again struggles with the second bottleneck due to insufficient failure feedback. As training progresses (c), subgoals selected by $\pi^h$ become more effective. The updated path refinement guides the agent through safer routes, enabling successful traversal of both bottlenecks and completion of the long-horizon task.

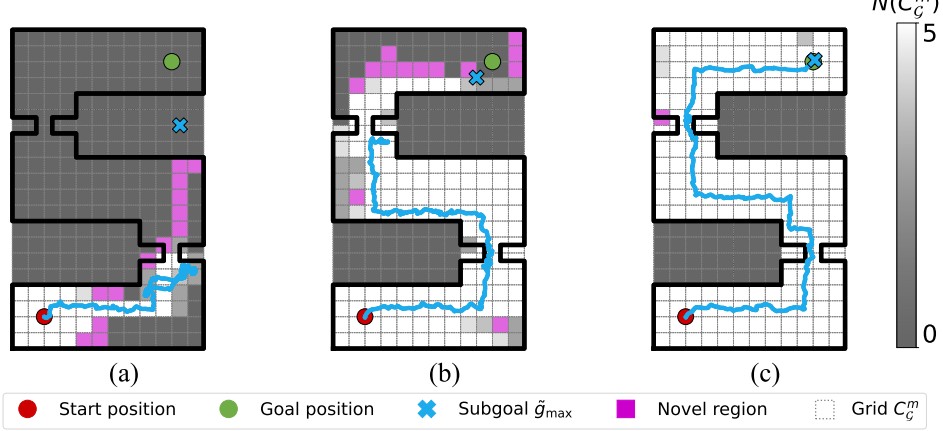

Figure 12: Trajectory analysis of SSE subgoals $\tilde{g}_{\max}$ in `AntMazeDoubleBottleneck` across: (a) early stage, (b) mid training, and (c) task success.

In Fig. 13, the agent in `AntKeyChest` begins by exploring the environment without a clear objective (a). Upon discovering the key (b), $\pi^h$ increasingly selects subgoals leading to the key location. In the final stage (c), the agent demonstrates the ability to sequentially reason over subtasks, first reaching the key and then navigating to the final goal with the required flag active, completing the task successfully.

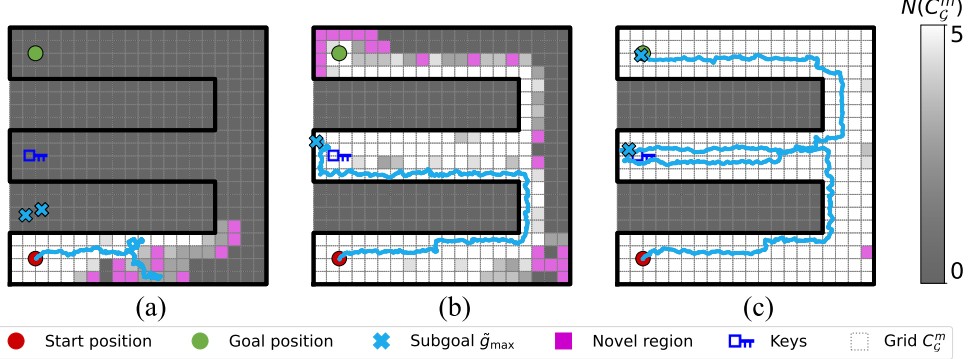

Figure 13: Trajectory analysis of SSE subgoals $\tilde{g}_{\max}$ in `AntKeyChest` across: (a) early stage, (b) after reaching the key, and (c) reaching the goal with the key (task success).

### E.2 ADDITIONAL ABLATION STUDIES

**Component Evaluation**

Fig. 14 presents the performance of SSE variants with key components ablated. We evaluate four variants: (1) SSE w/o $\mathcal{B}_F^h$ which replaces FER with a standard replay buffer, (2) SSE with HER which uses a conventional fixed-step policy, (3) SSE w/o $\pi^{\exp}$ which disables decoupled exploration, and (4) SSE w/o PR which omits path refinement. Removing path refinement degrades performance in long-horizon settings where mitigating unreliable transitions is crucial. Similarly, disabling the exploration policy consistently impairs goal-space coverage across all environments. Critically, SSE with HER and SSE w/o $\mathcal{B}_F^h$ fail in complex tasks like `AntDoubleKeyChest` despite achieving minor success in simpler maps. This failure demonstrates that conventional HER offers insufficient feasibility signals. Furthermore, it confirms that FER is vital for providing stop-on-failure feedback to teach the high-level policy to avoid unreliable subgoals. These components collectively ensure the decisive signals required for long-horizon planning.

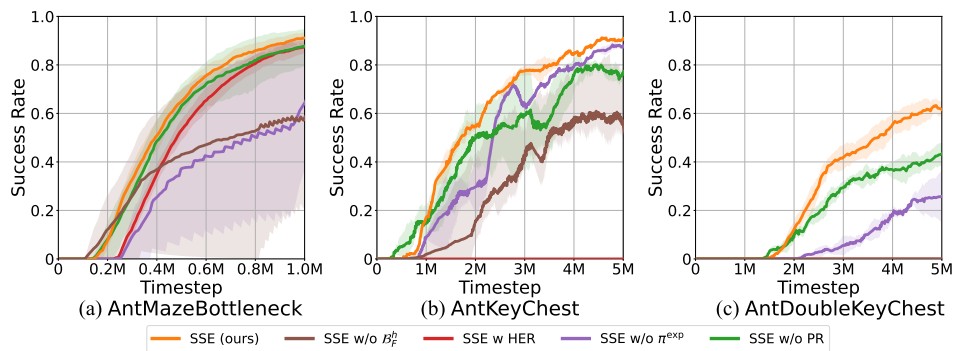

Figure 14: Component evaluation results in various maps

**Effect of Path Refinement Scaling Factor $c_{\text{dist}}$**

Fig. 15 shows how the scaling factor $c_{\text{dist}}$ affects performance. This parameter controls the trade-off between path efficiency and safety. In simpler tasks like `AntMazeBottleneck`, its impact is minimal. In more complex, long-horizon tasks, its role becomes more pronounced. A moderate

value, such as $c_{dist} = 5$, provides an effective balance, guiding the planner away from failure-prone regions without being overly conservative. In contrast, excessively high values (e.g., 10 or 20) can lead to inefficient detours, while a value of 1 may not sufficiently penalize risky paths. Overall, the performance remains high for values between 2 and 10, indicating a wide effective range.

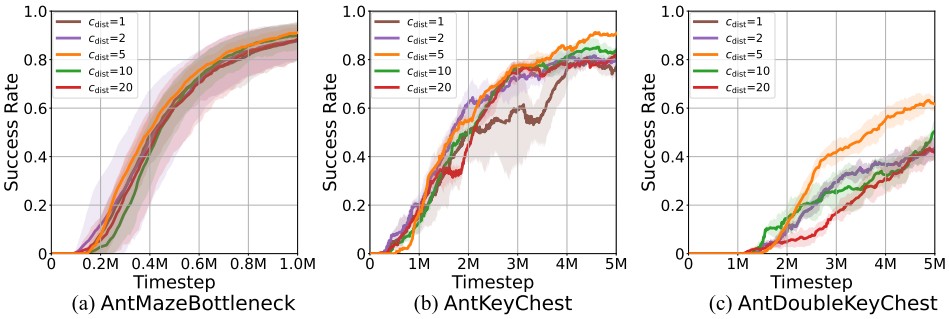

Figure 15: Distance scail $c_{dist}$ analysis in various maps

## Effect of Grid Size $d_{\mathcal{G}}$

Fig. 16 illustrates the effect of grid resolution $d_{\mathcal{G}}$. This parameter balances the granularity of novelty detection with the stability of failure statistics. The analysis shows that performance is not highly sensitive to this parameter within the tested range. For instance, in AntMaze environments, there is little significant difference in performance for $d_{\mathcal{G}}$ values of 1, 2, and 4. A coarse grid ($d_{\mathcal{G}} = 4$) may merge distinct regions, while an overly fine grid ($d_{\mathcal{G}} = 1$) can make failure statistics less reliable. Thus, we find that a moderate resolution ($d_{\mathcal{G}} = 2$ for AntMaze) provides a suitable balance. The optimal choice is dependent on the environment's scale, with smaller, structured spaces like Reacher benefiting from a finer grid ($d_{\mathcal{G}} = 1$).

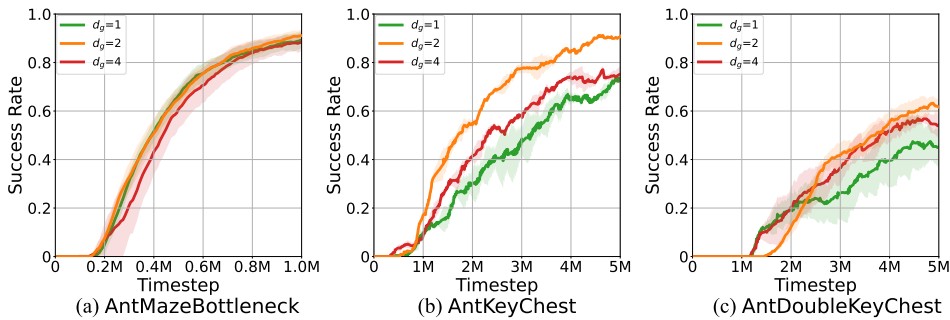

Figure 16: Grid size $d_{\mathcal{G}}$ analysis in various maps

## Effect of Exploration Ratio $\eta$

Fig. 17 shows how performance varies with the exploration ratio $\eta$, which controls the classic exploration-exploitation trade-off. In simple environments like `AntMazeBottleneck`, performance is consistent across a wide range of $\eta$ values. In long-horizon tasks like `AntDoubleKeyChest` that require discovering intermediate objectives, the influence of $\eta$ is naturally greater, but effective performance can be achieved with straightforward tuning. As shown in our experiments, setting the exploration ratio within a small range of 0.1 to 0.2 is sufficient to achieve strong performance across all tasks. This indicates that while the balance is important, extensive tuning of this hyperparameter is not required.

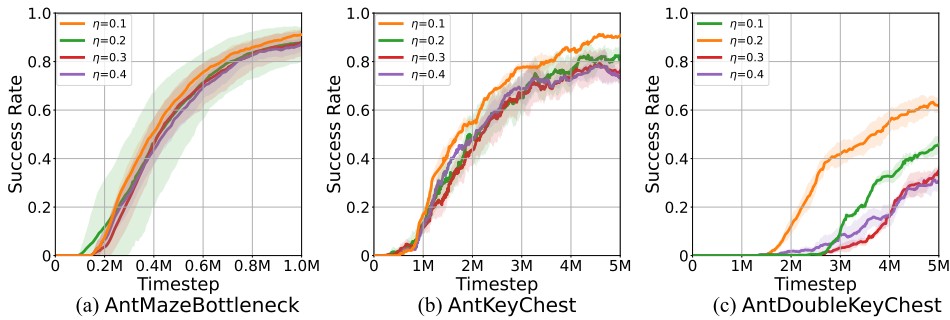

Figure 17: Exploration ratio $\eta$ analysis in various maps

**Effect of Reachability Check Threshold $\lambda$**

Fig. 18 illustrates the performance variation with respect to the reachability threshold $\lambda$. We employ the standard environment-provided value as the default. Performance remains stable as long as $\lambda$ is set smaller than the final task goal threshold, ensuring sufficient precision for intermediate waypoints. Conversely, extremely small values like 0.1 degrade performance in long-horizon tasks by forcing the low-level policy to achieve excessive precision. This hinders efficient exploration. These results confirm that the framework is robust to $\lambda$ and that the standard environment value offers an optimal balance.

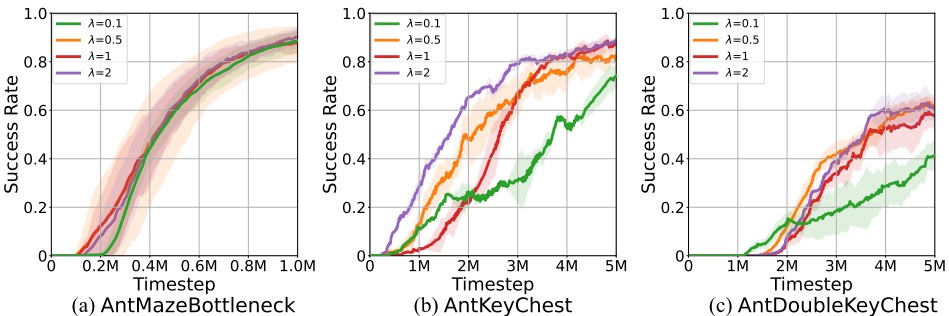

Figure 18: Distance threshold $\lambda$ analysis in various maps

**Effect of High-Level Gamma $\gamma^h$**    Fig. 19 confirms that $\gamma^h = 0.4$ yields optimal performance. Theoretically, SSE enforces strict, single-step reachability, which inherently condenses the high-level decision horizon. In this regime, a standard high discount factor (e.g., $\gamma^h = 0.99$) fails to distinguish between immediate and delayed successes. Conversely, a lower $\gamma^h$ sharpens credit assignment, making the value function $Q^h$ highly sensitive to temporal inefficiencies. This sensitivity guides the policy to prioritize direct, high-probability transitions, effectively delineating the reachability frontier and accelerating convergence.

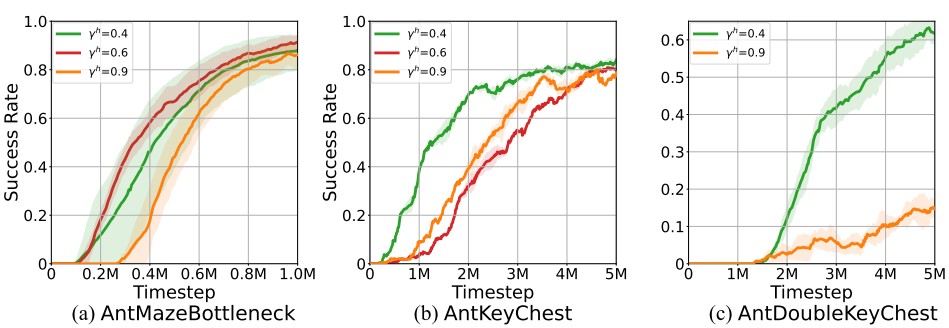

Figure 19: High-level gamma $\gamma^h$ analysis in various map

### E.3    COMPARISON ON ADDITIONAL ENVIRONMENTS

To assess generalizability, we conducted experiments on `AntPush`, `AntFall`, and `Ant4Rooms`. Since conventional baselines like HIRO failed to learn in these settings, we focus our comparison on the graph-based methods BEAG and NGTE. As shown in Fig. 20, standard graph-based algorithms struggle in tasks with complex contact dynamics or invisible obstacles. Their rigid reliance on shortest-path planning often leads to failure in environments like `AntPush` and `AntFall`. In contrast, SSE demonstrates superior performance. We attribute this to the decoupled exploration policy $\pi^{\text{exp}}$. This mechanism allows the high-level policy $\pi^h$ to leverage diverse exploration data beyond the graph's shortest path, facilitating robust planning under complex physical constraints. Note that in `AntPush`, we froze graph edge updates after initial convergence to mitigate instability from object interactions. SSE maintained its performance advantage even under this stabilization protocol.

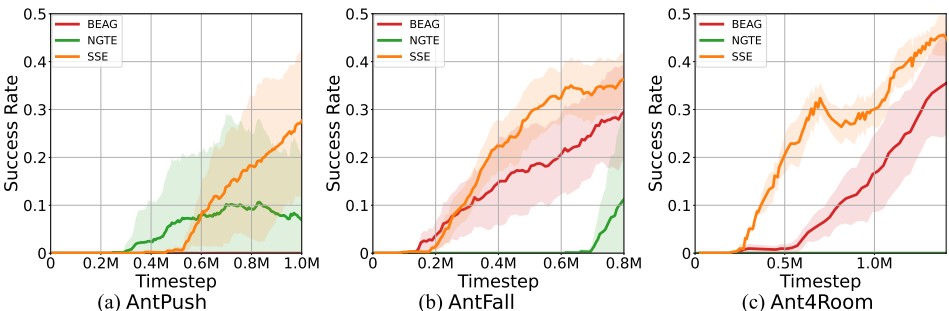

Figure 20: Performance comparison on additional environments

### E.4    MORE RANDOM SEEDS

To validate the statistical stability of our results, we supplemented our initial experiments (run with 5 seeds, a common practice in this field) by re-running the key complex tasks with 10 random seeds. We focused on tasks like AntKeyChest and AntDoubleKeyChest (Fig. 6, Fig. 8), where the standard deviation range of SSE with 5 seeds already showed a clear separation from the baselines.

As shown in Fig. 21, the experiment with 10 seeds produces a nearly identical performance curve, and no significant change in the mean success rate was observed compared to 5 seeds. More importantly, we re-affirmed that the standard deviation bounds remained clearly separated from the baseline algorithms. This confirms that the superior performance of SSE was not achieved by chance with a low seed count and that our findings are reliable and statistically stable.

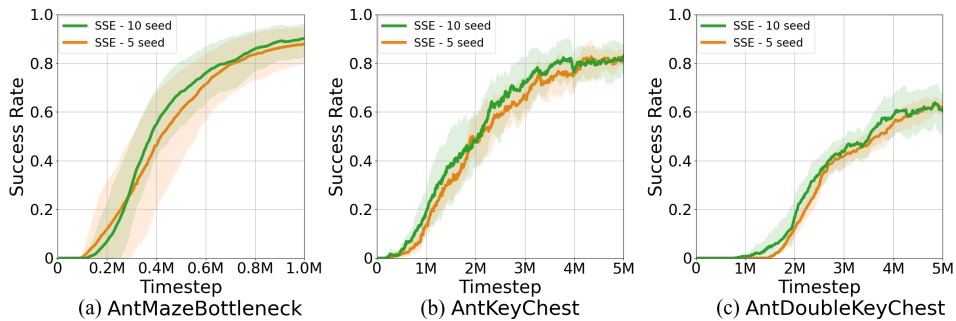

Figure 21: Performance comparison of SSE with 10 seeds

