# OpenReview forum: "Strict Subgoal Execution: Reliable Long-Horizon Planning in Hierarchical Reinforcement Learning"
_ICLR.cc/2026/Conference — ICLR 2026 Poster_

### Official Review · Reviewer_dYta · 2025-10-29

**Soundness:** 2
**Presentation:** 3
**Contribution:** 1
**Rating:** 2
**Confidence:** 4

**Summary:**

This paper proposed a novel method for hierarchical reinforcement learning, which leverages a frontier-based and failure-aware meta-policy replay buffer to ensure that goals that are reachable and at the frontier of experience are sampled. The method keeps track of this information with a graph-based encoding of the nodes in the environment. To ensure proper exploration, they leverage an exploration policy that is aware of all grid cells in 2D/3D space, and samples goals (at least partially) from these cells.

The main contribution are the novel replay buffer, and the method for having the graph be aware of what paths will fail, and having exploration be decoupled for goal space coverage.

**Strengths:**

- A key strength of their method is that it learns transitions between goals, enabling more efficient planning (as implied by Figure 1). It appears that they are effectively learning a high-level policy.
- The method achieves strong training performance compared to baseline approaches.
- The evaluations demonstrate that each component of the algorithm plays an essential role in achieving good performance in the environment.
- The experiments clearly illustrate the benefits of the proposed components (e.g., Figures 2 and 3).
- The overall presentation is clear and well-explained.

**Weaknesses:**

- "We assume the existence of a mapping φ such that φ(s) ∈ G, allowing the agent to infer goal progress from the current state."
  This seems like a strong assumption. Is this mapping learned, or predefined? Clarifying this would help understand how generalizable the approach is.

- The algorithm appears to involve substantial hand-crafting, which may limit its applicability to more complex or continuous environments. For instance, sampling from a grid-based estimator may not scale beyond grid-world settings.
  Similarly, assuming that subgoal reachability can be determined by $||\phi\left(s_{t^{\prime}}\right)-\tilde{g}_t||<\lambda$ is a strong and potentially unrealistic assumption.  Using this grid discretization for failure awareness also seems restrictive.

**Questions:**

- how is $\lambda$ set? I'm surprised there's no ablation on this.
- the Strict Subgoal Execution (SSE) framework updates the high-level policy with positive returns only when the low level successfully reaches the assigned subgoal. This seems like the same high-level ideas as [1]. Could you contrast to this work? This seems like a graph-based version of that idea?
- when comparing to methods like HIRO, what steps do you take to ensure that HIRO can use similar assumptions as this method, e.g. access to grid discretization scheme?
- isn't having exploration be decoupled from goal space coverage a common strategy in this graph-based planning setting, e.g. [2], which you did not cite.

[1] Self-Imitation Learning
[2] Successor feature landmarks for long-horizon goal-conditioned reinforcement learning

---

> ### Author Response · Authors · 2025-11-20
>
> We appreciate the reviewer dYta’s comments, which helped us refine both the presentation and the analysis. Below, we provide detailed responses to each question and clarify the corresponding revisions in the manuscript.
>
> **Weakness 1 ($\phi$ mapping assumption)**: Thank you for the question. As discussed in Section 2, we adopt **the standard graph-based HRL assumption** of a predefined mapping $\phi:\mathcal{S}\rightarrow\mathcal{G}$ that places each state in the goal space, so $\phi(s)\in\mathcal{G}$. This assumption is widely used because high-level subgoal selection and goal-conditioned evaluation require knowing the agent’s location in the goal space at each time step. Without $\phi$, the agent cannot compute goal proximity or success criteria in a goal-conditioned objective.
>
> **Weakness 2**: Thank you for the constructive feedback. We address the grid-based design and the subgoal reachability criterion separately
>
> **(Grid-based design)**: Regarding the grid-based design, while we initially prioritized it for simplicity in 2D/3D spaces, we agree that demonstrating scalability is crucial. Accordingly, in Sections 4.2 and 4.3 **we now provide, for both novelty-region estimation and failure-aware path refinement, not only the original grid-based formulation but also an explicit model-based variant, denoted SSE (Model), which is designed to scale to higher-dimensional goal spaces as below.**
>
> (i) Exploration policy: We revise Eq. 4 so that it accommodates both the grid-based and model-based approaches. In the model-based variant, we implement novelty using a prediction error signal by defining the novel region as $\lbrace\arg\max_{v\in V}\Vert f_{\eta}(v)-f_{\eta_{\mathrm{targ}}}(v) \rVert\rbrace$, where $f_{\eta_\mathrm{targ}}$, where $f_{\eta_\mathrm{targ}}$ is a randomly initialized target network and $f_\eta$ is a learned predictor. Since the predictor is trained to minimize this error on visited vertices, underexplored vertices maintain larger prediction error and are therefore selected as novel nodes for exploration.
>
> (ii) Failure-aware path refinement: Similarly, we revise Eq. 5 so that it covers both the grid-based and model-based approaches. In the model-based case, we train a failure prediction network $F_{\xi}$ with a cross-entropy loss to predict subgoal failure, labeling nodes on failed trajectories as $1$ and nodes on successful trajectories as $0$. As a result, $F_{\xi}(v)$ estimates the probability of failure at node $v$, which replaces the grid-based failure ratio in the refinement scheme.
>
> In Section 5.1, we present a comparative study of the two variants, where SSE (Model) nearly matches the performance of SSE (Grid), indicating that SSE is sufficiently flexible to extend to higher-dimensional goal spaces. However, SSE (Model) tends to be slightly slower than SSE (Grid) due to the training time of models, which suggests that the grid-based variant is preferable in simple 2D/3D goal spaces, while the model-based variant is more suitable in higher-dimensional settings. We appreciate the reviewer’s suggestion, and these additions clarify the scalability aspects of our method.
>
> **(Subgoal reachability)**: Regarding subgoal reachability, the check $\lVert \phi(s_{t'}) - \tilde{g}_t \rVert < \lambda$ is not an unrealistic assumption but the standard mechanism used in HRL to train the low-level policy. When the high level selects a subgoal, the low level must determine whether that subgoal has been reached. As shown in Eq. 8, $\pi^{l}$ receives a reward of $0$ or $-1$ depending on whether the waypoint $\mathrm{wp}_i$ is reached, which necessitates a threshold such as $\lambda$. **All graph-based HRL baselines we compare against adopt the same practice, including DHRL, NGTE, and BEAG**. The $\lambda$ values we use (Table 3) follow the same standard ranges. To avoid confusion, Section 2.2 has been revised to clarify the low-level reachability check and its role in our FER definition.

---

> ### Author Response · Authors · 2025-11-20
>
> **Question 1 (Reachability check threshold $\lambda$)**: As noted in our response to Weakness 2, $\lambda$ is the standard hyperparameter that determines whether the low-level policy has reached the high-level subgoal. We use the same values as prior work (DHRL, NGTE, BEAG) for the main results to ensure comparability. Since $\lambda$ is central to FER, we additionally report a sensitivity study in Appendix E.2. The ablation shows that SSE exhibits some variation with $\lambda$, but performance is not overly sensitive provided $\lambda$ is not set to extremes. Very small $\lambda$ makes the reachability check too strict, and very large $\lambda$ makes it uninformative. Within a broad intermediate range, SSE remains stable and effective. Reflecting the reviewer's feedback, we have revised Section 2.2 and the introduction of Section 5 to explicitly clarify this standard configuration used across all baselines. We also updated Section 5.2 to point readers to these additional sensitivity results.
>
> **Question 2 (Comparison to SIL)**: Thank you for the helpful suggestion. SIL and our SSE with FER differ in both objective and mechanism. SIL prioritizes trajectories that yielded higher returns in the past so that the agent reproduces them more often, which accelerates exploitation. If adapted to HRL, this emphasis on replaying successful behavior is conceptually closer to hindsight-style relabeling. In contrast, FER is designed to teach the high level where subgoals are actually achievable. As defined in Definition 4.1, FER records stop-on-failure transitions that assign zero return and terminate the episode at the failure point, and partial-success transitions that store the last reliably reached waypoint. These signals delineate the reachability frontier so that the high level avoids infeasible subgoals rather than imitating past successes that may be misleading for planning.
>
> In summary, SIL encourages the agent to imitate high-return successes to improve exploitation, while FER trains the agent to avoid failure-inducing choices in order to improve planning reliability. SSE is therefore not a graph-based variant of SIL. It is a replay mechanism tailored to high-level decision making in HRL that uses failure and partial-success information to shape the subgoal space into achievable regions.
>
> **Question 3 (Assumptions vs. HIRO)**: Thank you for the question. To be clear, SSE does not introduce assumptions beyond those standard in graph-based HRL, including HIRO. We assume a known goal space $\mathcal{G}$ for selecting high-level subgoals (our grid approach also uses this assumption), a mapping $\phi:\mathcal{S}\\!\to\\!\mathcal{G}$ to measure goal progress, and a reachability check threshold $\lambda$ for low-level completion. These are the same assumptions used across graph-based HRL baselines in GCRL settings.
>
> **Question 4 (Comparison to [2], SFL)**: Thank you for highlighting the relation to decoupled exploration. Methods like NGTE and the “successor feature landmarks” approach SFL primarily expand exploration near a graph’s frontier of already visited states, growing coverage outward from the current boundary. We included NGTE as a baseline for precisely this reason, and we add SFL to related works. Please note also that SFL targets state-as-goal settings that require a target image, whereas our evaluations use standard goal spaces without that constraint.
>
> Our use of the word “frontier” refers to a different concept. In FER, the frontier is the reachability boundary that separates subgoals the current low-level can reliably achieve from those it cannot. This boundary is established by stop-on-failure and partial-success transitions and is independent of where exploration happens to have reached. In short, NGTE/SFL focus on an exploration frontier on the graph, while FER defines a reachability frontier for high-level learning. Decoupled exploration in SSE is a supportive component to improve coverage, but the core novelty is FER’s reachability-based replay, which our ablations show is essential for stable high-level training.
>
> Once again, we thank the reviewer for their detailed feedback and insightful suggestions. We hope our clarifications help address the raised concerns and further highlight the contribution of our work.

---

### Official Review · Reviewer_BjYj · 2025-10-30

**Soundness:** 3
**Presentation:** 3
**Contribution:** 3
**Rating:** 6
**Confidence:** 3

**Summary:**

This paper introduces Strict Subgoal Execution (SSE), a hierarchical reinforcement learning framework designed for long-horizon, goal-conditioned tasks with sparse rewards. SSE incorporates Frontier Experience Replay (FER) to identify and filter out unreachable subgoals, improving the reliability and efficiency of high-level planning. It also uses a decoupled exploration policy to better explore the goal space and a path refinement mechanism to adjust edge costs based on low-level failures. Experiments on multiple benchmarks demonstrate that SSE achieves higher efficiency and success rates than existing goal-conditioned and hierarchical RL methods.

**Strengths:**

1. The manuscript is clearly written, with figures that effectively elucidate the proposed methodology and equations that comprehensively convey its technical details.

2. The experimental design is well structured, incorporating appropriate selections of baselines. The ablation study provides a thorough examination of the contribution of each component and offers a detailed analysis of the method’s sensitivity to hyperparameters.

**Weaknesses:**

1. To make a stronger case for the method's generality, the paper should include results from a broader set of environments. Specifically, in line with the existing HRL works, the authors may want to evaluate the method on Pusher, AntFall, AntGather and Ant4Rooms, in addition to the relatively simpler maze-based tasks. This would provide a better understanding of how well SSE adapts to various state spaces and task complexities. Including additional tasks would also help demonstrate whether the method can avoid unstable regions and enforce subgoal completion.

2. The use of only five random seeds in the comparative study may be insufficient to establish statistical significance. Increasing the number of seeds would lead to more reliable and robust experimental conclusions.

**Questions:**

Would it be possible to include comparative studies on a broader range of environments, using a larger number of random seeds as suggested?

---

> ### Author Response · Authors · 2025-11-20
>
> We appreciate the reviewer BjYj’s comments, which helped us refine both the presentation and the analysis. Below, we provide detailed responses to each question and clarify the corresponding revisions in the manuscript.
>
> **Weakness 1 (Environment diversity)**: Thank you for emphasizing the importance of evaluating SSE across a broad set of environments. We agree with this goal. The paper already reports results on nine challenging benchmarks, including newly introduced tasks such as AntMazeKeyChest and ReacherDoubleGoal, which are designed to stress long-horizon planning and subgoal reliability. Classic suites like AntPush, AntFall, and Ant4Room are valuable but are generally solved by prior methods, whereas our main evaluations focus on settings that better expose the weaknesses SSE is intended to address.
>
> In response to the reviewer's request, we conducted additional comparisons on AntPush, AntFall, and Ant4Room (AntGather was omitted because the public codebase has been removed). The new results, included in Appendix E.3, indeed highlight a key strength of our approach. These environments (e.g., AntPush, AntFall) expose a vulnerability in standard graph-based methods like BEAG or NGTE, whose strict reliance on the shortest path is ill-suited for complex dynamics. **The results show that SSE, while also graph-based, achieves markedly superior performance in these settings**. We attribute this to our decoupled exploration policy, which allows the high-level policy to leverage data from wider exploration instead of strictly adhering to the graph's shortest path. This leads to more effective planning, demonstrating a clear advantage over other graph-based methods in these specific tasks. These findings reinforce the effectiveness of SSE in complex interaction settings. We appreciate the suggestion and believe the added experiments improve the completeness of the evaluation.
>
> **Weakness 2 (Additional random seeds)**: Thank you for the thoughtful suggestion. Our primary results follow the common practice of using five random seeds, and in Fig. 6 the standard deviation bands for SSE are largely non-overlapping with those of competing methods, which suggests a comfortable margin. To further clarify statistical stability, we ran ten seeds on the key tasks and report the outcomes in Appendix E.4. The ten-seed learning curves closely track the five-seed results, indicating that our conclusions are stable and reproducible. In the final version we will present ten-seed results in the main text for completeness, and Appendix E.4 documents the side-by-side comparison between five and ten seeds.
>
> Once again, we thank the reviewer for their detailed feedback and insightful suggestions. We hope our clarifications help address the raised concerns and further highlight the contribution of our work.

---

### Official Review · Reviewer_kJYV · 2025-10-31

**Soundness:** 2
**Presentation:** 2
**Contribution:** 2
**Rating:** 4
**Confidence:** 2

**Summary:**

This paper has proposed a novel goal-conditioned hierarchical reinforcement learning approach, called SSE, to achieve reliable long-horizon planning. The proposed approach is evaluated in a set of simulated navigation tasks.

**Strengths:**

•  The paper addresses an important problem in hierarchical reinforcement learning — unreliable subgoal execution — which is crucial for long-horizon tasks.

•  The introduction of Frontier Experience Replay (FER) is conceptually clear and provides a principled way to delineate reachable and unreachable subgoals, improving training stability.

**Weaknesses:**

•  The proposed framework assumes that the goal space is known and low-dimensional, which may not hold for complex real-world manipulation or visual tasks where the goal representation itself is high-dimensional and uncertain.

•  The technical novelty is moderate — SSE combines known components (graph-based HRL, experience replay, path cost reweighting) rather than introducing fundamentally new learning principles.

•  All experiments are conducted in simulators; there is no validation in real-world robotic systems, limiting the practical credibility of the claimed reliability.

**Questions:**

1.	Does the proposed method assume that the goal space is low-dimensional? This may not be true for complex manipulation tasks.

2.	Is the proposed method applicable to real-world tasks? Only conducting experiments in simulators is not convincing enough.

---

> ### Author Response · Authors · 2025-11-20
>
> We sincerely thank Reviewer kJYV for the constructive feedback. Based on your comments, we have implemented a fully learning-based variant of our method to demonstrate scalability and clarified the novelty of our approach.
>
> **Weakness 1/Question 1(Goal space assumption)**: Thank you for the thoughtful comment. We address the goal-space assumption and the grid-based design separately. As noted in Section 2.2, **assuming a known goal space $\mathcal{G}$ is standard in graph-based HRL** because it defines the action domain for high-level subgoal selection and supports graph landmark construction. In practical settings, it is also reasonable to assume knowledge of the approximate range of goals targeted by the task.
>
> Regarding the grid, while we initially prioritized the grid-based estimator for its simplicity and efficiency in 2D/3D spaces, we agree that demonstrating scalability is crucial. Accordingly, in Sections 4.2 and 4.3 **we now provide, for both novelty-region estimation and failure-aware path refinement, not only the original grid-based formulation but also an explicit model-based variant, denoted SSE (Model), which is designed to scale to higher-dimensional goal spaces as below.**
>
> (i) Exploration policy: We revise Eq. 4 so that it accommodates both the grid-based and model-based approaches. In the model-based variant, we implement novelty using a prediction error signal by defining the novel region as $\lbrace\arg\max_{v\in V}\Vert f_{\eta}(v)-f_{\eta_{\mathrm{targ}}}(v) \rVert\rbrace$, where $f_{\eta_\mathrm{targ}}$ is a randomly initialized target network and $f_\eta$ is a learned predictor. Since the predictor is trained to minimize this error on visited vertices, underexplored vertices maintain larger prediction error and are therefore selected as novel nodes for exploration.
>
> (ii) Failure-aware path refinement: Similarly, we revise Eq. 5 so that it covers both the grid-based and model-based approaches. In the model-based case, we train a failure prediction network $F_{\xi}$ with a cross-entropy loss to predict subgoal failure, labeling nodes on failed trajectories as $1$ and nodes on successful trajectories as $0$. As a result, $F_{\xi}(v)$ estimates the probability of failure at node $v$, which replaces the grid-based failure ratio in the refinement scheme.
>
> In Section 5.1, we present a comparative study of the two variants, where SSE (Model) nearly matches the performance of SSE (Grid), indicating that SSE is sufficiently flexible to extend to higher-dimensional goal spaces. However, SSE (Model) tends to be slightly slower than SSE (Grid) due to the training time of models, which suggests that the grid-based variant is preferable in simple 2D/3D goal spaces, while the model-based variant is more suitable in higher-dimensional settings. We appreciate the reviewer’s suggestion, and these additions clarify the scalability aspects of our method.
>
> **Weakness 2 (Novelty)**: Thank you for the thoughtful feedback. The central novelty of our work is Frontier Experience Replay (FER), a replay principle designed specifically for high-level learning in HRL. We identify a failure mode of hindsight relabeling at the high level, where infeasible subgoals are relabeled as successes and planning is distorted. FER resolves this by adding two high-level transition types alongside success: stop-on-failure, which assigns zero return and terminates the episode at failure, and partial success, which records the last reliably reached waypoint. These signals mark the reachability frontier and reveal where failure occurs, so the high level learns to select achievable subgoals and avoids wasting high-level decisions. To our knowledge, **this success versus failure delineation at the high level has not been proposed before, and it is the key mechanism enabling SSE to operate reliably under non-stationary low-level skills**.
>
> We also study a failure-aware path refinement that increases the costs of graph edges with high observed failure rates, aligning high-level planning with what the low level can execute. Empirically, FER is essential and failure-aware refinement further improves performance. In our ablations, replacing FER with HER or removing path refinement causes substantial degradation or outright failure on long-horizon tasks, whereas the proposed SSE enables stable learning and strong performance, demonstrating that FER provides the critical learning signal while path refinement complements it by steering planning away from unreliable subgoals.

---

> > ### Author Response · Authors · 2025-11-27
> >
> > **Weakness 3/Question 2 (Simulation-based evaluation)**: Thank you for the thoughtful question. Our work concentrates on the efficiency and reliability of the core RL algorithm, and in this community it is customary to assess algorithmic utility and generalization on established simulation benchmarks rather than on real-robot platforms. Deploying on hardware introduces additional domain factors that are orthogonal to the algorithm itself, including sim-to-real transfer, hardware noise, etc. These issues constitute a substantial research area in robotics and fall outside the scope of this paper.
> >
> > In this paper, our aim is to provide a foundation that improves subgoal reliability in long-horizon GCRL tasks. The extensive simulation studies demonstrate that SSE is effective across diverse settings. Building on these results, evaluating SSE on real robotic systems is an important direction for future work, where the algorithm can be combined with system-level techniques to address sim-to-real challenges.
> >
> > We hope our clarifications help address the raised concerns and further highlight the contribution of our work.

---

### Official Review · Reviewer_cF88 · 2025-11-01

**Soundness:** 3
**Presentation:** 3
**Contribution:** 3
**Rating:** 8
**Confidence:** 3

**Summary:**

The paper mentioned that in long-horizon, sparse-reward tasks, high-level policies often choose subgoals that the low-level controller can’t reliably reach. When HER is applied at the high level, failures get relabeled. The authors propose Frontier Experience Replay (FER), which stores three kinds of high-level transitions—success, stop-on-failure (zero-return, early termination), and partial success to the last reliably reached waypoint. They introduce a decoupled exploration policy that prioritizes under-explored goal-space regions (simple grid-density estimator) alongside an ε-greedy high-level policy to improve coverage. The authors add failure-aware path refinement that inflates edge costs in high-failure regions of the goal graph, nudging Dijkstra planning away from unstable corridors. SSE substantially outperforms HRL (HIRO, HRAC) and graph-based methods (HIGL, DHRL, NGTE, PIG, BEAG) on success rate and often on learning speed

**Strengths:**

Strong empirical results on a diverse suite, including tasks that require implicit sequencing
Ablation coverage is thoughtful: removing FER or replacing with HER largely breaks performance on harder tasks

**Weaknesses:**

The density estimator and failure statistics hinge on a grid. This is fine for 2D/3D but will be problematic in higher dimensions and for goals that include orientation or other factors.
No guarantees or formal properties regarding convergence or bias introduced by early termination/FER.

**Questions:**

This is a strong empirical paper with a simple, well-motivated idea that addresses a real failure mode in hierarchical goal-conditioned RL. Have you thought about how to extend to analyze theoretically with HER?
How often does path refinement genuinely alter the planned path (e.g., fraction of episodes where the refined path differs from shortest path)?
have you tried replacing grid-based novelty with learned density?if so, do the gains persist?

---

> ### Author Response · Authors · 2025-11-20
>
> We appreciate the reviewer cF88’s comments, which helped us refine both the presentation and the analysis. Below, we provide detailed responses to each question and clarify the corresponding revisions in the manuscript.
>
> **Weakness 1 (Extension to higher-dimensional goal spaces)**:
>
> Thank you for raising this important point. Thank you for raising this important point. We agree that demonstrating scalability beyond grid-based settings is crucial. Accordingly, in Sections 4.2 and 4.3 **we now provide, for both novelty-region estimation and failure-aware path refinement, not only the original grid-based formulation but also an explicit model-based variant, denoted SSE (Model), which is designed to scale to higher-dimensional goal spaces as below.**
>
> (i) Exploration policy: We revise Eq. 4 so that it accommodates both the grid-based and model-based approaches. In the model-based variant, we implement novelty using a prediction error signal by defining the novel region as $\lbrace\arg\max_{v\in V}\Vert f_{\eta}(v)-f_{\eta_{\mathrm{targ}}}(v) \rVert\rbrace$, where $f_{\eta_\mathrm{targ}}$ is a randomly initialized target network and $f_{\eta}$ is a learned predictor. Since the predictor is trained to minimize this error on visited vertices, underexplored vertices maintain larger prediction error and are therefore selected as novel nodes for exploration.
>
> (ii) Failure-aware path refinement: Similarly, we revise Eq. 5 so that it covers both the grid-based and model-based approaches. In the model-based case, we train a failure prediction network $F_{\xi}$ with a cross-entropy loss to predict subgoal failure, labeling nodes on failed trajectories as $1$ and nodes on successful trajectories as $0$. As a result, $F_{\xi}(v)$ estimates the probability of failure at node $v$, which replaces the grid-based failure ratio in the refinement scheme.
>
> In Section 5.1, we present a comparative study of the two variants, where SSE (Model) nearly matches the performance of SSE (Grid), indicating that SSE is sufficiently flexible to extend to higher-dimensional goal spaces. However, SSE (Model) tends to be slightly slower than SSE (Grid) due to the training time of models, which suggests that the grid-based variant is preferable in simple 2D/3D goal spaces, while the model-based variant is more suitable in higher-dimensional settings. We appreciate the reviewer’s suggestion, and these additions clarify the scalability aspects of our method.
>
> **Weakness 2/Question 1 (Analysis of FER)**: Thank you for the thoughtful suggestion. Our work focuses on a practical failure mode in HRL: unstable subgoal reachability at the low level. We introduce SSE, FER, and early termination as pragmatic mechanisms tailored to this setting, and we center the paper on their empirical effectiveness. A formal convergence analysis is challenging here because the high level learns in a non-stationary environment. The dynamics and rewards perceived by $\pi^h$ evolve with the low-level policy, so the effective MDP changes over time. Standard analyses often rely on assumptions that are misaligned with this setting, such as perfect subgoal completion by the low level, which is precisely the phenomenon we aim to address.
>
> Instead, **we provide empirical evidence for the effect of FER and its advantage over HER**. In the component evaluation of Fig. 8, removing FER (w/o $\mathcal{B}\_F^h$) or substituting HER (SSE w/ HER) induces unnecessary subgoal generation and substantially degrades performance, whereas FER supports stable learning and state-of-the-art results. The trajectory analysis in Appendix E.1 further shows that subgoal sequences under FER stabilize training by filtering unreachable subgoals and concentrating updates on achievable ones. A full convergence analysis under coupled high- and low-level learning is an important direction beyond the scope of this paper, and we view it as valuable future work.
>
>
> **Question 2 (Fraction of path refinement)**: Thank you for the question. Path refinement in our method is a structural reweighting of graph edges rather than an episodewise tweak. Once a region is sufficiently visited and shows a high failure rate, the costs of its incident edges are increased, and all subsequent episodes plan on this refined graph fixed before each episode. Consequently, the “fraction of episodes” whose path changes equals the proportion whose unrefined shortest path would have traversed refined edges. In bottlenecked layouts such as AntmazeBottleneck, failures concentrate near wall-adjacent corridors and this fraction is typically high, whereas in open layouts with fewer constrictions it is lower.
>
> Once again, we thank the reviewer for their detailed feedback and insightful suggestions. We hope our clarifications help address the raised concerns and further highlight the contribution of our work.

---

### Author Response · Authors · 2025-11-20

We sincerely thank all reviewers for their constructive feedback. Following your suggestions, we have substantially strengthened the manuscript with additional experiments and analyses to better demonstrate the scalability and reliability of our framework. A revised version, with all changes highlighted in blue, has been uploaded. The major updates are summarized below.

**(i) Model-based implementation for scalability to higher-dimensional goal spaces (Sections 4.2, 4.3, and 5)**: To demonstrate that SSE is scalable beyond grid settings, we revised the methodology to explicitly define two variants: the original grid-based approach, denoted SSE (Grid), and a new model-based approach, denoted SSE (Model). The latter implements exploration and path refinement using neural estimators, achieving performance comparable to the grid-based implementation. This supports the feasibility of extending our framework to higher-dimensional goal spaces.

**(ii) Clearer problem setup and expanded related work (Sections 2, 3, and 5)**: We clarified that our setup aligns with the standard assumptions and interfaces adopted by prior HRL works, including the state-to-goal mapping $\phi$, the low-level reward design, and the subgoal reachability criterion. We also broadened the related-work discussion to situate our contributions more precisely.

**(iii) Broader environments and additional ablations (Section 5, and Appendix E.2, E.3, E.4)**: In response to the request for diversity, we added evaluations on physically challenging tasks with complex contacts (AntPush, AntFall, and Ant4Rooms) and included further ablations, such as increasing the number of random seeds and a sensitivity study over the reachability check threshold $\lambda$. These additions strengthen the empirical support for our claims.

We believe these revisions address the main concerns raised during review and improve the clarity and completeness of the paper. We are grateful for the reviewers’ guidance, which materially enhanced the manuscript.

**On Nov. 27**: During the rebuttal period, we further revised the manuscript so that the model-based approach for covering higher-dimensional goal spaces, which multiple reviewers pointed out as important, is now described in the main-text rather than only in the appendix. We have also updated the corresponding responses accordingly and kindly ask the reviewers to take these changes into consideration.

---

### Author Response · Authors · 2025-12-02

Dear Area Chair,

Thank you very much for taking the time to review our work. In addition to the per-reviewer responses and common comments below, we would like to briefly summarize the main contribution of our proposed SSE method and how the rebuttal process proceeded, in the hope that this helps your assessment.

---

**Main contribution:** Our proposed SSE method targets goal-conditioned RL in long-horizon tasks where reaching the final goal is particularly challenging. Existing graph-based hierarchical RL methods typically apply hindsight experience replay to both high-level and low-level policies, often leading the high-level policy to treat subgoals as successful even when they are not accurately reached. This induces instability and repeated selection of unnecessary subgoals, making complex tasks hard to solve. SSE addresses this by introducing **Frontier Experience Replay (FER), which explicitly separates truly failed parts of a trajectory from partially successful regions, enabling the agent to avoid failure regions**. The high-level policy receives a reward only when the subgoal is reached accurately, reducing spurious successes and ensuring efficient learning. On top of this, **a failure-aware path refinement module steers the agent away from regions that frequently cause failure, leading to more reliable paths**. As a result, SSE is, to our knowledge, the only method that successfully learns in complex multi-goal environments such as KeyChest and DoubleGoal, where prior methods fail, indicating a strong contribution to the GCRL domain.

---


**Summary of rebuttal:** From the reviewers’ comments, our understanding is that most reviewers agree that SSE makes a meaningful contribution and shows strong empirical performance. The main concerns were not about the core algorithm, but about assumptions on the goal space, especially scalability to higher-dimensional settings (reviewers *cF88*, *kJYV*, *dYta*), as well as clarification of the setup and requests for further experiments and analysis (reviewers *cF88*, *BjYj*, *dYta*). In response, we focused on **extending SSE beyond the grid-based approach: in Sections 4 and 5, we reformulated the exploration policy and failure-aware path refinement so that they also admit a model-based instantiation (Eqs. 4 and 5) to handle higher-dimensional goal spaces.** This led us to define two variants, SSE (Grid) and SSE (Model), and we provided experiments and analysis for both: their performance is very similar, suggesting that one can choose either variant, which we believe directly addresses the scalability concern. We also clarified that **our setting does not rely on any additional assumptions beyond those in prior graph-based HRL methods**, and added experiments on AntPush, AntFall, and Ant4Rooms, together with hyperparameter and random-seed ablations in Appendix E, all of which directly address the requested analyses and consistently show advantages for SSE.

---


We find it very unfortunate that, due to the OpenReview system issue, reviewers were unable to post follow-up comments. However, the reviewers who raised the largest number of concerns (*kJYV* and *dYta*) both pointed to the scalability of the grid-based approach and perceived extra assumptions as their primary issues, and, as outlined above, we believe these points have been sufficiently resolved by introducing SSE (Model) and clarifying that our assumptions match those of existing graph-based HRL baselines. Reviewer *cF88* also discussed scalability to higher-dimensional goal spaces but did not raise similar concerns about our treatment of the goal space and already gave a positive score, suggesting that the other reviewers might also lean more positive once these clarifications are taken into account. The other reviewer, *BjYj*, was already positive and mainly requested additional experiments and analysis; for each of these points, we provided detailed responses and incorporated the corresponding results into the revised manuscript, so we expect that their overall assessments would have remained positive.

---


In summary, we believe that SSE offers clear novelty and a distinct contribution. During the rebuttal period we resolved the most important concern regarding higher-dimensional goal spaces, both at the derivation level and experimentally, and the extended experiments and ablations further strengthened the paper and addressed the main points raised by the reviewers. We hope this overview is helpful for your decision, and we are sincerely grateful for your careful consideration of our work.

---

### Meta-Review · Area_Chair_xJDE · 2026-01-07

**Summary:**

The paper introduces strict subgoal execution (SSE), a graph-based hierarchical reinforcement learning framework that incorporates frontier experience replay to improve long-horizon, goal-conditioned learning. By explicitly filtering infeasible subgoals and leveraging failure information to refine high-level plans, SSE significantly improves learning efficiency and task success rates compared to prior methods.

Overall, based on both the paper and the review–rebuttal discussion, I believe this work makes a meaningful and well-justified contribution to hierarchical RL. The technical ideas are sound, the empirical results are compelling, and the paper is clearly written. I therefore recommend acceptance of this paper.

**Reviewer Concerns:**

In the rebuttal, the authors clarify that the main reviewer concerns were largely about scalability and assumptions on the goal space, rather than weaknesses in the core algorithm. These concerns are convincingly addressed through a model-based variant of SSE that extends beyond grid-based environments. The additional experiments, ablation studies, and clarifications demonstrate that SSE adheres to assumptions comparable to existing approaches, scales to higher-dimensional tasks, and consistently outperforms strong baselines.

**Reviewer Scores:**

See above comments.

---

### Decision · Program_Chairs · 2026-01-26

Accept (Poster)